# EGFR transactivates RON to drive oncogenic crosstalk

Carolina Franco Nitta[1], Ellen W Green[1], Elton D Jhamba[1], Justine M Keth[1], Iraís Ortiz-Caraveo[1], Rachel M Grattan[1], David J Schodt[2], Aubrey C Gibson[1], Ashwani Rajput[3,4], Keith A Lidke[2,4], Bridget S Wilson[1,4], Mara P Steinkamp[1,4], Diane S Lidke[1,4]*

[1]Department of Pathology, University of New Mexico, Albuquerque, United States; [2]Department of Physics & Astronomy, University of New Mexico, Albuquerque, United States; [3]Department of Surgery, University of New Mexico, Albuquerque, United States; [4]Comprehensive Cancer Center, University of New Mexico, Albuquerque, United States

**Abstract** Crosstalk between different receptor tyrosine kinases (RTKs) is thought to drive oncogenic signaling and allow therapeutic escape. EGFR and RON are two such RTKs from different subfamilies, which engage in crosstalk through unknown mechanisms. We combined high-resolution imaging with biochemical and mutational studies to ask how EGFR and RON communicate. EGF stimulation promotes EGFR-dependent phosphorylation of RON, but ligand stimulation of RON does not trigger EGFR phosphorylation – arguing that crosstalk is unidirectional. Nanoscale imaging reveals association of EGFR and RON in common plasma membrane microdomains. Two-color single particle tracking captured formation of complexes between RON and EGF-bound EGFR. Our results further show that RON is a substrate for EGFR kinase, and that transactivation of RON requires formation of a signaling competent EGFR dimer. These results support a role for direct EGFR/RON interactions in propagating crosstalk, such that EGF-stimulated EGFR phosphorylates RON to activate RON-directed signaling.

*For correspondence: dlidke@salud.unm.edu

Competing interest: The authors declare that no competing interests exist.

## Editor's evaluation

The study by Nitta et al., brings a sophisticated understanding of the mechanisms behind crosstalk between two mitogenic growth factor receptor tyrosine kinases – EGFR and RON. While earlier studies indicated that the receptors interact, this work provides evidence that the EGFR-RON crosstalk is unidirectional. They also show that this interaction takes place specifically at the membrane, rule out other intermediary molecules and locations between EGFR and RON. Consistent with this they find that, in vitro, RON acts as a substrate of the EGFR kinase domain. The study therefore identifies many mechanistic details about this particular interaction that could be relevant from a clinical standpoint to develop better drugs that can reduce the oncogenic potential of these receptors. From a fundamental biology standpoint, the study reveals novel mechanisms of signal transduction at the membrane.

## Introduction

There is growing evidence demonstrating that crosstalk between members of distinct receptor tyrosine kinase (RTK) subfamilies can drive tumorigenesis and therapeutic resistance. Understanding these complicated interactions is critical for the development of novel dual-targeting therapeutics to improve patient outcomes (*Arteaga, 2007*; *Bardelli et al., 2013*; *Choudhary et al., 2016*; *Engelman*

*et al., 2007*; *Follenzi et al., 2000*; *Hsu et al., 2006*; *Lai et al., 2009*; *Prahallad and Bernards, 2016*). Here, we focus on the coordinated signaling between the Epidermal Growth Factor Receptor (EGFR, the canonical member of the EGFR/ErbB/HER subfamily) and Recepteur d'Origine Nantais (RON, also known as MST1R and a member of the MET subfamily). Prior evidence has implicated EGFR/ RON crosstalk in the modulation of important cellular responses, notably migration and invasiveness in cancer (*Keller et al., 2013*; *Maggiora et al., 2003*; *Yao et al., 2013*). RON expression combined with EGFR correlates with poorer outcomes for cancer patients. In head and neck cancer, EGFR/RON co-expression is associated with decreased event-free survival, while in bladder cancer, co-expression correlates with increased tumor invasion, increased recurrence after first-line therapy, and decreased patient survival (*Hsu et al., 2006*; *Keller et al., 2013*). Direct interactions between RON and EGFR have been inferred from co-immunoprecipitation studies (*Hsu et al., 2006*; *Peace et al., 2003*), as well as observations that EGFR/RON complexes can translocate into the nucleus to act as transcription factors (*Liu et al., 2010*). These previous studies demonstrate EGFR/RON crosstalk, but do not provide details on the nature of the interaction between the receptors that can be used to understand mechanism.

Since the extracellular domains of EGFR and RON are so structurally distinct, it is difficult to explain their interactions through traditional dimerization models (*Chao et al., 2012*; *Ogiso et al., 2002*). For EGFR, ligand binding introduces structural rearrangements that promote dimerization and kinase activity. These include rotation of the extracellular domain exposing the dimerization arm to stabilize receptor dimers (*Burgess et al., 2003*; *Chung et al., 2010*; *Freed et al., 2017*; *Low-Nam et al., 2011*; *Valley et al., 2015*), dimerization of the transmembrane domains, formation of helical dimers between the juxtamembrane domains (*Jura et al., 2009*), and asymmetric orientation of the kinase domains that allows for allosteric activation (*Zhang et al., 2006*). While EGFR has been shown to form ligand-independent dimers, the shorter lived interactions and maintenance of the autoinhibitory mechanisms prevents these short-lived dimers from initiating signaling (*Chung et al., 2010*; *Jura et al., 2009*; *Low-Nam et al., 2011*; *Valley et al., 2015*; *Yu et al., 2002*). Although the mechanisms of RON activation and potential dimerization are not as well studied, crystallographic studies of the RON extracellular domain have suggested that RON homodimers can form in the absence of ligand (*Chao et al., 2012*).

Here, we combined high-resolution imaging with rigorous biochemical measurements to dissect the mechanisms underlying EGFR/RON crosstalk and to understand the nature of their interactions. We provide evidence of unidirectional crosstalk between EGFR and RON. Activation of EGFR by EGF leads to RON phosphorylation via direct phosphorylation of RON by EGFR's integral kinase, which is then further enhanced by RON's own catalytic activity. Importantly, EGFR activator or receiver mutants are incapable of promoting RON phosphorylation, demonstrating that RON cannot substitute for either partner of the EGFR asymmetric dimer. Taken together, our results support a molecular mechanism for crosstalk where RON, independent of its ligand MSP, acts as a co-receptor for EGF-bound EGFR dimers to promote RON activation and support RON-directed signaling outcomes.

## Results

### Generation of human cell lines co-expressing full-length RON and EGFR

We introduced full-length RON into two well-characterized human cell lines, A431 and HEK-293, to generate model systems. A431 squamous carcinoma cells have high levels of endogenous EGFR expression, and provide a model for tumors with high EGFR expression and modest levels of RON. HEK-293 human embryonic kidney cells have negligible levels of endogenous EGFR or RON, and provide a test bed for balanced expression of combinations of RON plus either wildtype or mutated forms of EGFR. The low levels of endogenous RON expression in these cell lines allowed us to stably express full-length HA-tagged RON (A431[RON] and HEK[RON]), while avoiding potential complications from endogenous alternatively spliced RON isoforms (*Bardella et al., 2004*; *Chen et al., 2000*; *Collesi et al., 1996*; *Wang et al., 2000*; *Zhou et al., 2003*). ACP-tagged EGFR was also stably introduced into HEK[RON] cells to generate a HEK-293 cell line expressing comparable levels of EGFR and RON (HEK[RON/EGFR]). Expression levels were evaluated by flow cytometry for both cell models. A431[RON] cells display ~2.2 million EGFR molecules and only ~92,000 RON receptors on the cell surface (~24:1

EGFR:RON ratio), whereas HEK^RON/EGFR cells express EGFR and RON at a ratio of ~2:1 (~600,000 EGFR; ~275,000 RON).

## Crosstalk between EGFR and RON is EGF-driven

We evaluated EGFR/RON crosstalk based on changes in receptor phosphorylation in response to each of their cognate ligands. EGF treatment led to the expected EGFR phosphorylation in both A431^RON and HEK^RON/EGFR cells (*Figure 1A*). MSP treatment induced RON phosphorylation in both cell lines (*Figure 1B*). Importantly, whereas MSP did not activate EGFR, treatment of cells with EGF promoted robust phosphorylation of RON (*Figure 1A and B*). This effect was dose-dependent and detectable at doses of EGF as low as 2 nM (*Figure 1—figure supplement 1*). In contrast, neither physiological levels (2–5 nM) nor high doses (50 nM) of MSP could induce EGFR phosphorylation at PY1068 or other EGFR phospho-tyrosine sites (*Figure 1—figure supplements 2 and 3*). This was the first indication that crosstalk is unidirectional in our two model systems, with crosstalk occurring from EGF-bound EGFR to RON but not from MSP-bound RON to EGFR. Note that our western blots resolved the mature RON (bottom RON band) from the pro-form (upper band; see *Figure 1—figure supplement 4*).

Dual stimulation with EGF and MSP did not increase EGFR phosphorylation beyond EGF alone (*Figure 1C*). However, combining EGF and MSP led to a synergistic enhancement in RON phosphorylation that is higher than expected from the additive effects of either ligand alone (*Figure 1D*). These results further support the conclusion that crosstalk occurs between full-length RON and EGFR, is unidirectional, and is EGF-driven. EGFR was often detected in RON immunoprecipitates, in both resting and stimulated cells, as a band co-migrating with pro-RON at 180 kDa via western blot analysis (using EGFR or EGFR-PY1068 antibodies) or identified by mass spectrometry, (*Figure 1—figure supplement 1* and *Supplementary file 1*). Co-immunoprecipitation of RON and EGFR in unstimulated cells has been reported previously (*Hsu et al., 2006*; *Peace et al., 2003*). In contrast to that previous work, we do not observe an increase in co-precipitation with ligand stimulation. However, we note that co-IP was not always evident, suggesting weak interactions, and our experiments were performed at earlier time points (5 min) than the previous studies (30 min).

## EGF induces similar phosphorylation kinetics for EGFR and RON

We next evaluated the early phosphorylation kinetics of RON and EGFR in response to physiological levels of ligand, either 5 nM MSP or 5 nM EGF. EGF-induced EGFR-PY1068 phosphorylation was rapid, peaking by 1 min (*Figure 1E*; top left blot and green line), as previously demonstrated (*Hsieh et al., 2010*; *Kovacs et al., 2015a*). RON phosphorylation after EGF treatment was similarly rapid, again reaching maximum phosphorylation levels by 1–2 min (*Figure 1E*; bottom left blot and blue line). In contrast, RON phosphorylation in response to MSP was slower, peaking at 2 min or later (*Figure 1E*; right blot and magenta line). The faster kinetics of EGF-driven RON phosphorylation when compared to MSP-driven RON phosphorylation may be a result of the higher affinity of EGF for EGFR (*Kauder et al., 2013*; *Lemmon, 2009*). However, the closely aligned EGF-induced EGFR and RON phosphorylation kinetics led us to postulate that RON is a substrate and co-receptor for the EGF-activated EGFR kinase.

## RON and EGFR co-cluster in plasma membrane nanodomains

Considering the rapid (< 5 min) time scale of crosstalk, we considered that EGF-induced phosphorylation of RON must be occurring at the plasma membrane. Given that we found crosstalk to be EGF-dependent, we focused on comparing receptor distributions in resting and EGF-stimulated cells. As a first step, we confirmed that RON and EGFR have similar distributions on the plasma membrane of HEK^RON/EGFR (*Figure 2A*) and A431^RON cells (*Figure 2—figure supplement 1*) using confocal microscopy.

We also applied our established transmission electron microcopy (TEM) technique with immunogold-labeled membrane sheets (*Yang et al., 2007*) to evaluate the nano-organization of RON with respect to EGFR. Receptor spatial distributions were determined from resting or EGF-stimulated A431^RON cells and imaged by TEM (*Figure 2B*). TEM images show that RON and EGFR frequently co-reside in mixed clusters in untreated cells (circles, *Figure 2B*, left panels). The co-clustering of the two receptors on resting membranes was confirmed by Ripley's K co-variant statistical test (*Wilson et al., 2004*; *Yang et al., 2007*; *Figure 2B*, bottom panels). EGFR/RON co-clustering was maintained after 2 min and 5 min of treatment with 50 nM EGF (*Figure 2B*, middle and right panels). While EM results

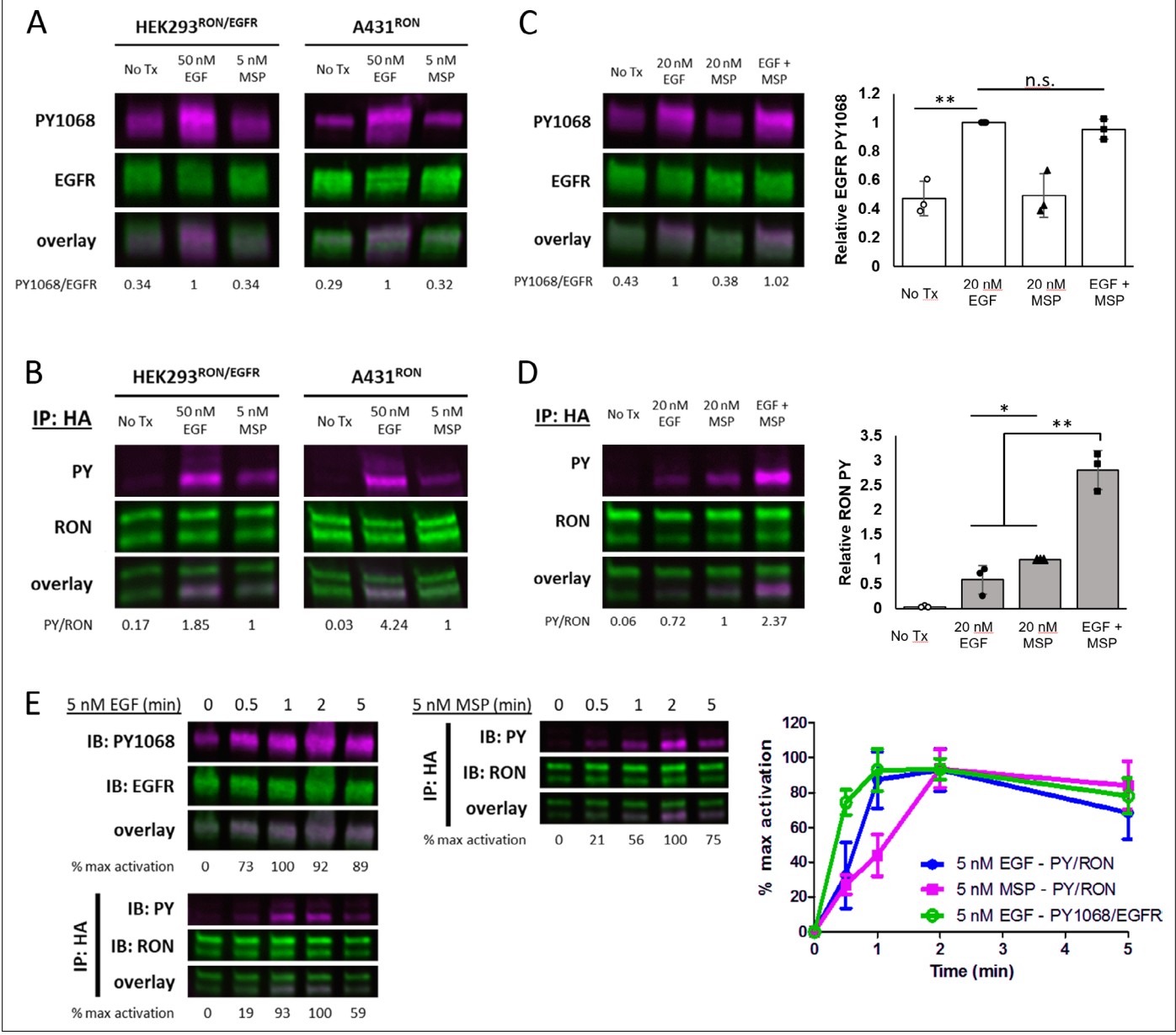

**Figure 1.** Crosstalk between EGFR and RON is EGF-driven. (**A and B**) HEK[RON/EGFR] or A431[RON] cells were treated with ± 5 nM MSP or 50 nM EGF for 5 min at 37 °C. Representative immunoblots showing PY1068 and EGFR on cell lysates (**A**) or pan-phosphotyrosine (PY) and RON on samples immunoprecipitated (IP) with anti-HA antibody (**B**). (**C and D**) A431[RON] cells were stimulated with ± 20 nM EGF, 20 nM MSP or both for 5 min at 37 °C and immunoblotted as in (**A** and **B**). Triplicate biological experiments are quantified in the bar graphs to the right, shown as mean ± SD. (**E**) Representative immunoblots of a phosphorylation time course for A431[RON] cells treated with 5 nM EGF or 5 nM MSP and immunoblotted as in (**A** and **B**). Graphed values (right) are from triplicate biological experiments, normalized to maximal activation, and presented as mean ± SD. * p < 0.05; ** p < 0.01.

The online version of this article includes the following source data and figure supplement(s) for figure 1:

**Source data 1.** Full raw western blots and blots with relevant bands labeled, corresponding to *Figure 1A, B, C, D and E*.

**Source data 2.** Source data for quantification of blots in *Figure 1C, D, and E*.

**Figure supplement 1.** A431[RON] cells were treated with increasing levels of EGF for 5 min.

**Figure supplement 1—source data 1.** Full raw western blots and blots with relevant bands labelled, corresponding to *Figure 1—figure supplement 1*.

**Figure supplement 2.** A431[RON] cells were treated with increasing levels of MSP for 5 min.

**Figure supplement 2—source data 1.** Full raw western blots and blots with relevant bands labeled, corresponding to *Figure 1—figure supplement 2*.

*Figure 1 continued on next page*

*Figure 1 continued*

**Figure supplement 3.** HEK^RON/EGFR cells were treated± EGF or MSP for 5 min.

**Figure supplement 3—source data 1.** Full raw western blots and blots with relevant bands labeled, corresponding to *Figure 1—figure supplement 3*.

**Figure supplement 3—source data 2.** Source data for quantification of blots in *Figure 1—figure supplement 3*.

**Figure supplement 4.** HEK^RON cells transiently transfected with EGFR-WT or EGFR-Δ998± EGF treatment for 5 min.

**Figure supplement 4—source data 1.** Full raw western blots and blots with relevant bands labeled, corresponding to *Figure 1—figure supplement 4*.

demonstrate co-clustering of these molecules, the static EM image cannot reveal whether or not the receptors are physically interacting or merely co-confined. Taken together with the observation that co-immunoprecipitation occurs in the absence of ligand, these data suggest that pre-existing protein complexes may be key contributors in EGFR-to-RON crosstalk.

## Crosstalk occurs at the plasma membrane

Given their co-localization at the plasma membrane and the rapid (< 5 min) unidirectional crosstalk discussed above, we hypothesized that RON and EGFR form hetero-oligomeric complexes to alter EGF-driven signaling output. Using single particle tracking (SPT) of Quantum Dot (QD)-labeled receptors, we evaluated the mobility of HA-RON on the surface of live A431^RON cells using a monovalent anti-HA Fab fragment conjugated to QD probes (QD605-HA-RON) (*Valley et al., 2015*). Previous work by ourselves and others has shown that mobility is a read-out for receptor phosphorylation status, such that a shift to slower mobility is correlated with receptor dimerization, signaling, and subsequent recruitment of downstream signaling molecules and/or signaling-induced alterations in the local environment (*Chung et al., 2010*; *Erasmus et al., 2016*; *Low-Nam et al., 2011*). *Figure 3A* shows the mean squared displacement (MSD) versus time lag (Δt) for tracking of QD605-HA-RON under different stimulation conditions. The distribution of Diffusion Coefficients (D) for individual cells is shown in *Figure 3B*. Consistent with ligand-induced phosphorylation and/or oligomerization, we observed that RON mobility is decreased upon MSP stimulation (*Figure 3A, B* and *Figure 3—figure supplement 1*). Notably, RON mobility is also decreased with EGF addition (*Figure 3A, B* and *Figure 3—figure supplement 1*). This EGF-induced mobility change was prevented when cells were treated with an EGFR kinase inhibitor (PD153035) (*Figure 3A, B*). In *Figure 3C and D*, confocal images show the location of RON and EGFR in HEK^RON/EGFR cells after 10 min of EGF stimulation. As expected, EGF-bound EGFR is rapidly endocytosed and shows obvious co-localization with the early endosome marker, EEA1. In contrast, RON receptors are not readily found in the early endosomes, and co-endocytosis of EGFR and RON within endosomes is rare (see *Figure 3—figure supplement 2* for quantification). The lack of RON co-endocytosis is further supported by TEM images from A431^RON cells, where EGFR, but not RON, was found in clathrin-coated pits 5 min after EGF addition (*Figure 3E*). These results suggest that EGFR/RON interactions are either sufficiently transient that EGFR is sorted for endocytosis, while RON remains on the surface, or that EGFR complexed with RON is retained longer on the cell surface. These data support the premise that EGFR-mediated activation of RON occurs rapidly at the plasma membrane, rather than in endosomes, and is dependent on EGFR kinase activity.

## EGF-bound EGFR and RON engage in direct interactions

To confirm that EGFR and RON interact at the cell membrane, we used simultaneous two-color QD tracking that allows direct detection and quantification of protein-protein interactions on live cells, as we have described previously (*Erasmus et al., 2016*; *Low-Nam et al., 2011*; *Steinkamp et al., 2014*; *Valley et al., 2015*). *Figure 4* demonstrates the visualization of receptor interactions by tracking of individual receptors in spectrally distinct channels at high spatiotemporal resolution. QDs were conjugated to either a monovalent anti-HA Fab fragment (*Steinkamp et al., 2014*; *Valley et al., 2015*) for RON (QD-HA-RON) or to EGF (*Lidke et al., 2004*; *Low-Nam et al., 2011*) to follow ligand-bound EGFR (QD-EGF-EGFR). We monitored RON/RON homo-interactions in A431^RON cells by labeling receptors with a mixture of anti-HA-QD605 and anti-HA-QD655 (*Figure 4A*). *Figure 4B* and *Figure 4—video 1* shows an example of a long-lived interaction between two QD-tagged RON receptors lasting for ~5 s

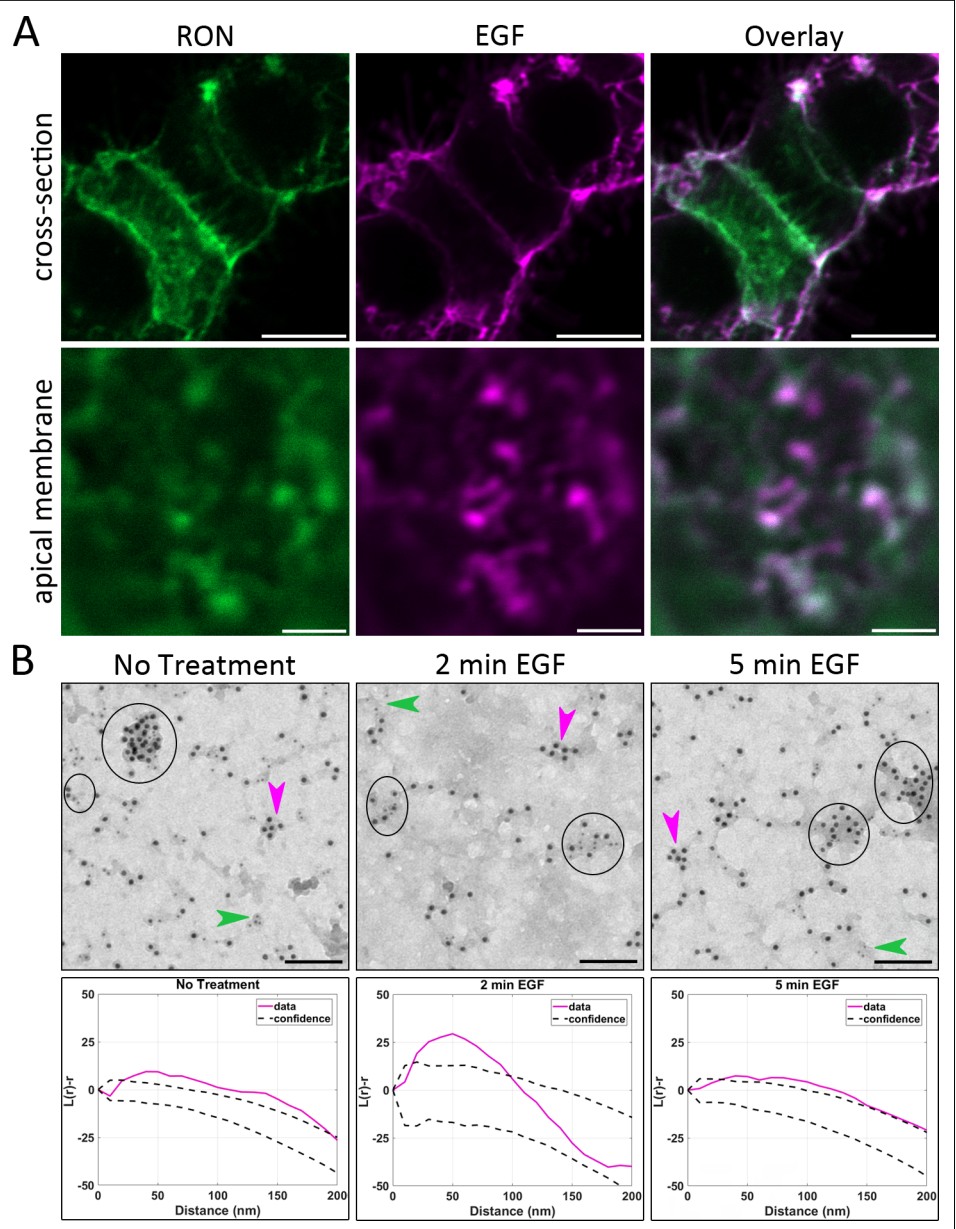

**Figure 2.** RON and EGFR co-cluster in plasma membrane nanodomains. (**A**) HEK[RON/EGFR] cells were first labeled for RON using α-HA-FITC Fab fragment (green), treated with 10 nM EGF-AF647 (magenta) for 5 min on ice and then fixed. Representative images from three biological replicates show colocalization of RON and EGFR at the plasma membrane. Scale bars, 10 µm (cross-section) and 2 µm (apical membrane). (**B**) Top row: Membrane sheets were prepared from A431[RON] cells ± 50 nM EGF for 2 and 5 min. Sheets were labeled on the cytoplasmic face using antibodies to RON (6 nm gold) and EGFR (12 nm gold). Circles indicate co-clusters of RON and EGFR in representative images from three biological replicates; arrowheads indicate clusters containing RON (green) or EGFR (magenta) only. Scale bar, 100 nm. Bottom row: Ripley's K bivariant function was used to evaluate co-clustering. The experimental values for L(r)-r (corresponding to EM image directly above) are shown in magenta and the 99 % confidence window for complete spatial randomness is plotted as dashed lines. In each case, experimental values are seen to fall above the confidence window, indicating co-clustering.

The online version of this article includes the following source data and figure supplement(s) for figure 2:

**Figure supplement 1.** Comparison of RON and EGFR colocalization using fluorescent EGF or anti-EGFR (R1, Santa Cruz) antibody in A431[RON] cells.

**Figure supplement 1—source data 1.** Source data for colocalization analysis in *Figure 2—figure supplement 1*.

*Figure 2 continued on next page*

*Figure 2 continued*

**Figure supplement 2.** Additional EM Analysis.

**Figure supplement 2—source data 1.** Source data for EM quantification in *Figure 2—figure supplement 2*.

before breaking apart. A range of dimer lifetimes was observed, and additional examples and videos of RON homo-interactions are found in *Figure 4—figure supplements 1–4*. Two-color tracking was next used to determine if RON and EGFR form hetero-complexes. Here, HA-RON was tracked using anti-HA-QD655 and endogenous EGFR was tracked using QD605-EGF (*Figure 4D*). This live cell imaging approach directly captures pairs of QD-labeled RON and EGF-bound EGFR that engage as complexes and move with correlated motion on the cell membrane. The example in *Figure 4E* and *Figure 4—video 2* shows a more transient interaction with a duration of ~1.5 sec (see further examples and videos in *Figure 4—videos 1–6*).

Quantification of correlated motion between receptors confirmed the formation of bona fide receptor complexes (*Low-Nam et al., 2011*). The presence of correlated motion was assessed over the full data set of the two-color trajectories (*Figure 4C and F*), reporting on the behavior of the overall population. Correlated motion was observed when two RON receptors were in close proximity, as indicated by the reduction in the uncorrelated jump distance at small separation seen in *Figure 4C*. Jump magnitude also decreases at small separation, indicating that RON homo-complexes are moving more slowly than monomers. Importantly, correlated motion is also clearly observed for RON and EGF-bound EGFR, confirming direct interactions between these disparate receptors (*Figure 4F*).

Using a two-state hidden Markov model (HMM) similar to that described in Low-Nam et al (*Low-Nam et al., 2011*), we estimated the dimerization kinetics between interacting receptors. In the absence of ligand, we found an off-rate ($k_{off}$) for RON/RON homo-interactions of $0.18 \pm 0.02$ s$^{-1}$ (average lifetime of ~5.5 s). Together with the correlated motion analysis, these results are consistent with the idea that RON can homodimerize independent of ligand, as was proposed by others based on the crystal structure of the RON extracellular domain (*Chao et al., 2012*) and the evidence for ligand-independent activation with RON overexpression or mutations in cancer (*Liu et al., 2011*; *Santoro et al., 1998*; *Wang et al., 2007*). Two-color tracking of QD655-HA-RON and QD605-EGF-EGFR returned an off-rate of $0.49 \pm 0.05$ s$^{-1}$ for hetero-interactions. This more transient (average lifetime of ~2 s) interaction is consistent with the ability of EGFR to phosphorylate RON without subsequent co-endocytosis. The cellular environment, including the availability of binding partners and ligand, may influence dimer stability. We note that the experiments described here are performed at low QD-EGF concentration and the frequency of interactions and off-rates may be altered with higher ligand dose or changes in receptor expression.

## Maximal EGF-induced RON phosphorylation requires kinase activity of both receptors

Treatment of A431$^{RON}$ cells with the reversible EGFR-selective kinase inhibitor, PD153035, blocks EGF-induced changes in RON mobility (*Figure 3A, B*). To follow-up these results implicating EGFR kinase activity as the primary driver of EGFR/RON crosstalk, we treated both A431$^{RON}$ and HEK$^{RON/EGFR}$ cells with the irreversible pan-ErbB kinase inhibitor, afatinib. Afatinib treatment completely blocks EGF-dependent phosphorylation of EGFR (*Figure 5A*) and RON (*Figure 5B*), but does not inhibit MSP-dependent RON phosphorylation (*Figure 5B*). Cells pretreated with BMS777607, a RON/Met-family kinase inhibitor, blocked MSP-dependent RON phosphorylation, but only partially blocked EGF-dependent RON phosphorylation (*Figure 5B*). As expected, BMS777607 did not affect EGF-dependent EGFR phosphorylation. These results indicate that both EGFR and RON kinase activity contribute to EGF-mediated RON phosphorylation.

To confirm the differential contributions of the EGFR and RON kinases in crosstalk, we expressed the kinase dead mutant of RON (RON-K1114M) in A431 cells (*Figure 5—figure supplements 1 and 2*). EGF-driven phosphorylation of RON-K1114M was observed and afatinib treatment abrogated this phosphorylation (*Figure 5—figure supplement 1*). The reduction in RON phosphorylation by BMS777607, as seen in RON-WT, is not observed for RON-K1114M since this mutant inherently lacks kinase activity. Consistent with the observed phosphorylation, HA-RON-K1114M undergoes significant

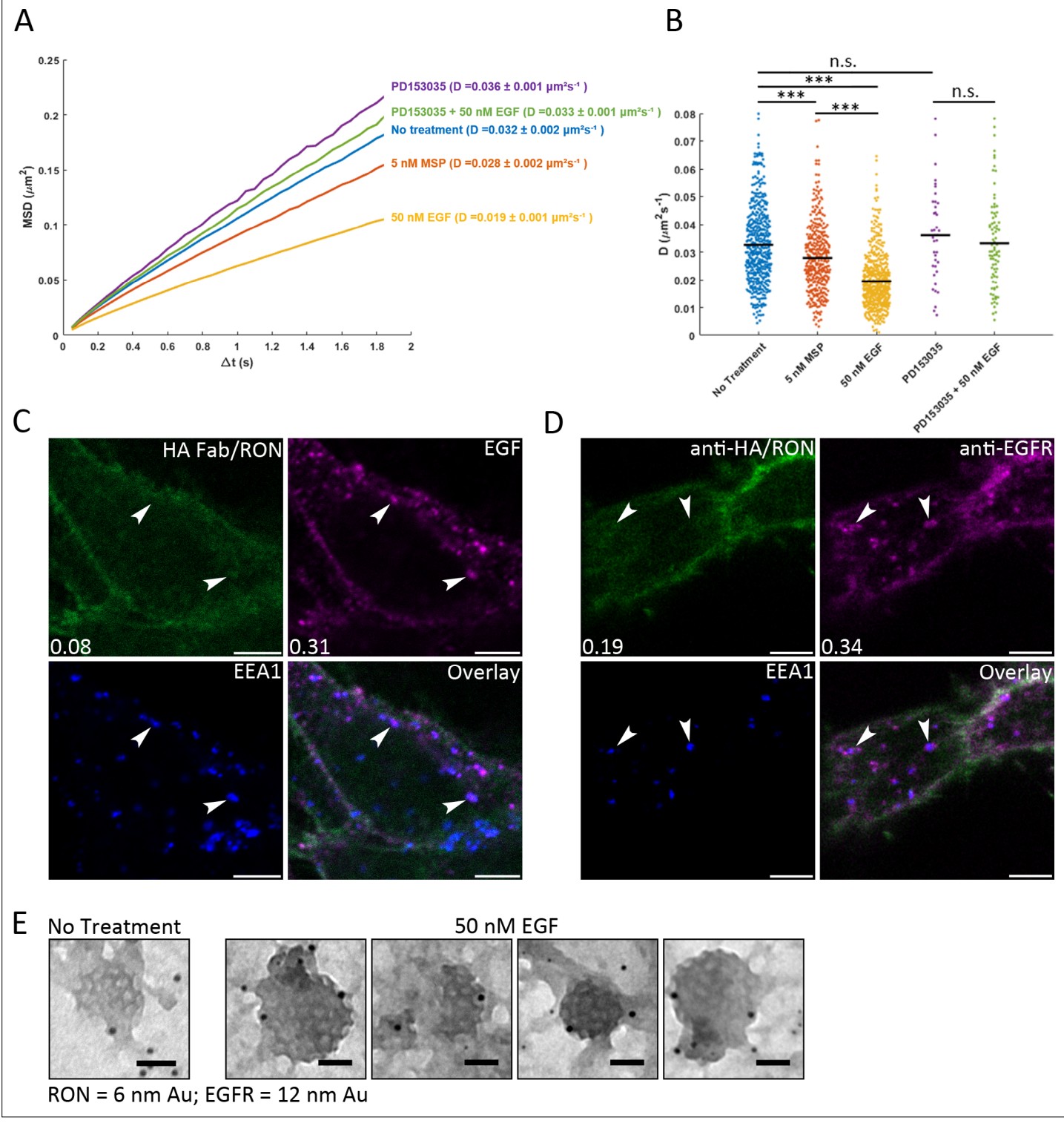

**Figure 3.** Crosstalk occurs at the plasma membrane. (**A**) Single particle tracking of QD605-HA-RON was used to quantify RON mobility on A431[RON] cells ± ligand. Ensemble mean squared displacement (MSD) shows reduction in slope of the MSD with ligand stimulation, indicating a reduced mobility. Treatment with EGFR kinase inhibitor prevents RON slow down with EGF. The number of jumps fit for each condition range from 42,183 to 898,300. (**B**) Corresponding distribution of diffusion coefficients, D, for individual cells is plotted for a range of 39 to 517 cells per condition; *** p < 0.001. (**C**) HEK[RON/EGFR] cells were labeled for RON with anti-HA-FITC Fab fragment (green), treated with 10 nM EGF-AF647 (magenta) for 5 min on ice followed by 10 min at 37°C, then fixed and labeled with an antibody to EEA1 (early endosomes, blue). Representative images from three biological replicates show that EGF-positive endosomes (arrows) primarily do not contain RON. Pearson's coefficient for the image shown and colocalization with EEA1 is shown in the

*Figure 3 continued on next page*

*Figure 3 continued*

bottom left corner. (**D**) Alternative labeling method for monitoring endosome content where HEK^RON/EGFR cells were treated with 50 nM EGF for 10 min at 37°C, fixed and then antibodies were used to label RON (anti-HA, green) or EGFR (magenta). Further quantification for C, D is in *Figure 3—figure supplement 2*. (**E**) Membrane sheets prepared from A431^RON cells ± 50 nM EGF for 5 min were labeled for RON (6 nm gold) or EGFR (12 nm gold). TEM images show clathrin-coated pit lattices on the cell membranes containing EGFR, but not RON. Scale bars, 50 nm.

The online version of this article includes the following source data and figure supplement(s) for figure 3:

**Source data 1.** Source data for diffusion coefficient distributions in *Figure 3B*.

**Figure supplement 1.** RON mobility is reduced in response to bothlow/physiological and high/saturating doses of ligand.

**Figure supplement 1—source data 1.** Source data for diffusion coefficient distributions in *Figure 3—figure supplement 1*, panel B.

**Figure supplement 2.** EGFR is more readily found in EEA1-positive early endosomes than RON after EGF stimulation.

**Figure supplement 2—source data 1.** Source data for colocalization analysis in *Figure 3—figure supplement 2*.

reduction in mobility with EGF stimulation in SPT experiments (*Figure 5—figure supplement 2*). RON's family member Met has been shown to transphosphorylate RON, as well as engage in crosstalk with EGFR (*Harwardt et al., 2020*; *Jo et al., 2000*). However, our results show that EGF-induced phosphorylation of HA-RON-K1114M is not reduced in the presence of BMS777607, indicating that Met is not involved in EGFR/RON crosstalk. These results underscore the importance of EGFR kinase activity in crosstalk and rule out Met as a possible contributor.

## EGFR/RON crosstalk does not require downstream signaling molecules

Thus far, our data indicate the critical role for EGFR kinase activity in EGF-dependent RON phosphorylation. While this could be attributed to direct phosphorylation of RON by EGFR in hetero-oligomeric complexes, an alternative mechanism could involve recruitment of EGFR-associated kinases such as the tyrosine kinase Src (*Danilkovitch-Miagkova et al., 2000*; *Sato et al., 1995*). To rule out the possibility that Src is an intermediary in propagating EGFR/RON crosstalk, A431^RON cells were pretreated with the Src family kinase inhibitor dasatinib prior to stimulation with 50 nM EGF. Low doses of dasatinib (10 nM) were used to ensure Src family specificity (*Nam et al., 2005*) while achieving 70% reduction in basal Src PY416 phosphorylation (*Figure 6—figure supplement 1*). Dasatinib treatment did not alter EGF-induced RON phosphorylation (*Figure 6A*), arguing that EGFR/RON crosstalk does not depend on Src kinase activity, and is likely to reflect direct action of the EGFR kinase (unaffected by dasatinib) on RON.

In addition to Src, EGFR also recruits a number of other cytoplasmic signaling molecules to phosphotyrosines in its C-terminal tail. We expressed in HEK^RON cells a version of EGFR truncated at amino acid 998 (HEK^RON/EGFR-Δ998), which lacks most of the phosphotyrosine binding sites that recruit downstream adaptor molecules (*Kovacs et al., 2015a*). In a previous study, EGFR-Δ998 exhibited decreased phosphorylation of the remaining tyrosine residues 845, 974, and 992 compared to full length EGFR suggesting that phosphorylation at these sites might depend on downstream binding partners (*Kovacs et al., 2015a*). Unexpectedly, stimulating HEK^RON/EGFR-Δ998 cells with EGF led to enhanced phosphorylation of RON compared to HEK^RON/EGFR-WT (*Figure 6B*). We speculated that the EGFR tail might compete for phosphorylation by the kinase domain, explaining why its deletion enhances RON phosphorylation.

These results confirm that recruitment of downstream signaling molecules to the C-terminal tail of EGFR is not required for EGF-driven RON phosphorylation, while raising a new question as to the mechanism of this enhanced crosstalk. We considered the possibility that truncation of the EGFR tail could prevent recruitment of EGFR-associated phosphatases that normally dampens downstream signals (*Kleiman et al., 2011*; *McCabe Pryor et al., 2015*). HEK^RON cells with EGFR-WT or EGFR-Δ998 treated with EGF followed by afatinib (to irreversibly inhibit subsequent rounds of phosphorylation) were examined for RON and EGFR phosphorylation (*Figure 6—figure supplement 2*). Independent of full-length or truncated EGFR, RON lacked phosphorylation after 20 s of afatinib treatment, confirming that the dephosphorylation kinetics are similar. Thus, while a third-party signaling molecule is not required to mediate crosstalk in our model systems, the unstructured EGFR tail or its binding partners appear to have a role in limiting EGFR-mediated phosphorylation of RON.

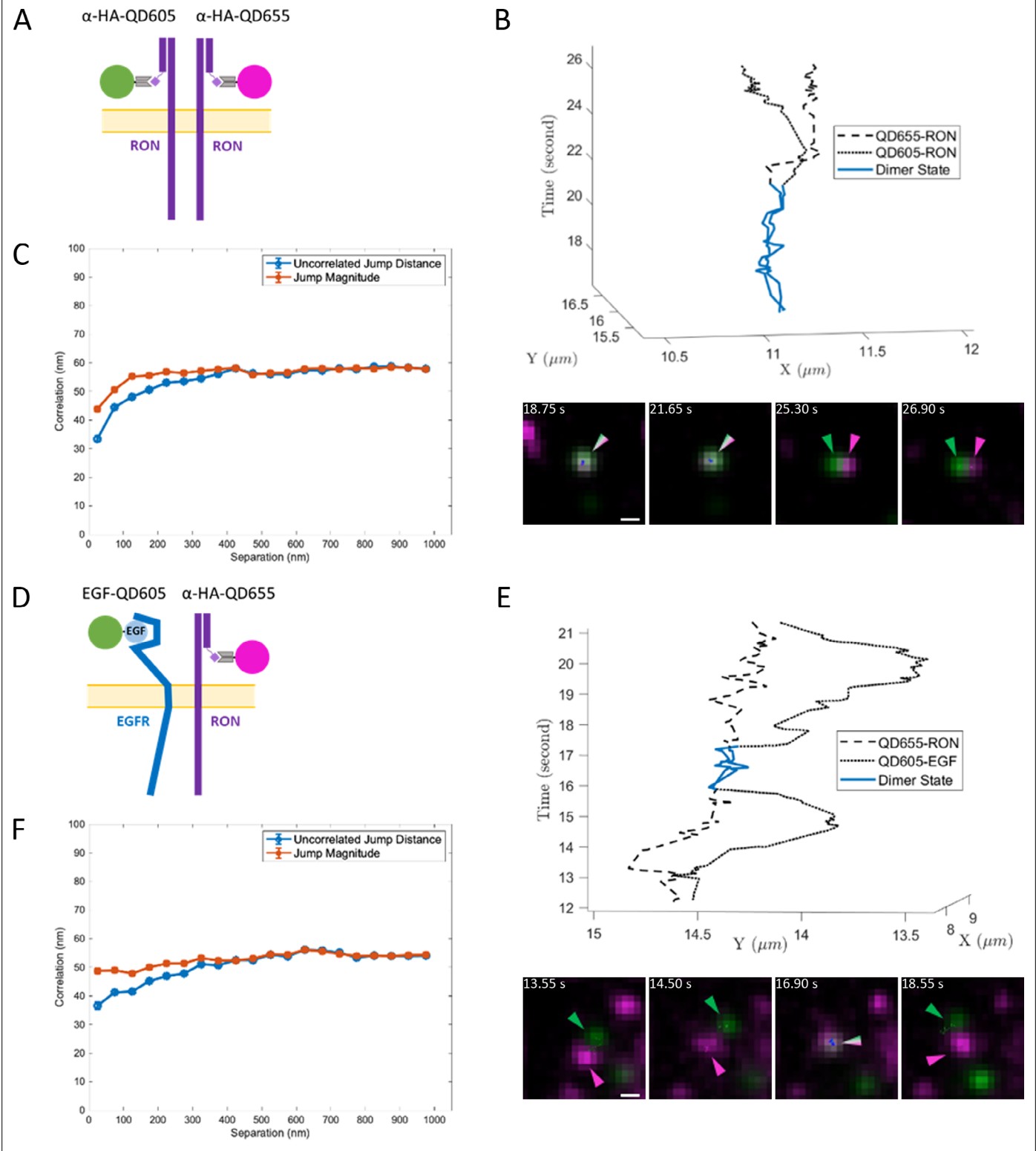

**Figure 4.** Two-color single QD tracking captures interactions between RON and EGFR. Two color SPT results for resting RON receptor interactions (**A–C**) and ligand-bound EGFR interactions with RON (**D–F**). (**A**) Schematic representation of two-color (anti-HA-QD605 and anti-HA-QD655) RON SPT. (**B**) Representative 3D trajectory (top) and time series (bottom) for a RON homo-interaction lasting ~5 s (blue) with accompanying *Figure 4—video 1*. Scale bar, 500 nm. (**C**) Ensemble correlated motion plot for all two-color RON tracking. The number of jumps for each data point range from 2,068–15,649. (**D**) Schematic representation of two-color SPT of EGF-bound EGFR (QD655-EGF) and RON (anti-HA-QD605). (**E**) Sample 3D trajectory (top) and time series (bottom) showing EGF-EGFR and RON interacting for ~1.5 s (blue) with accompanying *Figure 4—video 2*. Scale bar, 500 nm. (**F**) Ensemble correlated

*Figure 4 continued on next page*

*Figure 4 continued*

motion plot for all EGF-EGFR and RON tracking. The number of jumps for each data point range from 1,500–16,794.

The online version of this article includes the following video and figure supplement(s) for figure 4:

**Figure supplement 1.** Example 3D trajectory (top) of two QD-RON receptors that engaged in repeated transient interactions (blue segments).

**Figure supplement 2.** Example 3D trajectory (top) of a long-lived (~22 s) interaction between two QD-RON receptors.

**Figure supplement 3.** Example 3D trajectory (top) of EGF-bound EGFR and RON receptors that are initially found in a dimer complex (blue) that then dissociates at 36.5 s.

**Figure supplement 4.** Example 3D trajectory (top) of a long-lived (~16 s) interaction between EGF-bound EGFR and RON receptors.

**Figure 4—video 1.** Two RON receptors are engaged in an interaction from the start of the video, which lasts for ~5 s before the receptors dissociate.
https://elifesciences.org/articles/63678/figures#fig4video1

**Figure 4—video 2.** A short-lived interaction between QD605-EGF-EGFR (green) and QD655-RON (magenta).
https://elifesciences.org/articles/63678/figures#fig4video2

**Figure 4—video 3.** Two RON receptors undergoing repeated interactions that each last 1–2 s.
https://elifesciences.org/articles/63678/figures#fig4video3

**Figure 4—video 4.** A long-lived RON/RON interaction.
https://elifesciences.org/articles/63678/figures#fig4video4

**Figure 4—video 5.** A complex of EGF-EGFR and RON is seen to break apart after a ~ 3.5 s dimer event.
https://elifesciences.org/articles/63678/figures#fig4video5

**Figure 4—video 6.** A long-lived interaction between EGF-EGFR and RON.
https://elifesciences.org/articles/63678/figures#fig4video6

## RON is a substrate for EGFR kinase activity

Having ruled out a role for downstream signaling molecules, we hypothesized that the RON C-terminal tail is a substrate for EGFR kinase activity. To further test this possibility, we designed an in vitro kinase assay to allow for detection of EGFR phosphorylation of RON without background from other cellular components. In these experiments, we used immunoprecipitated kinase dead RON (RON-K1114M) as a substrate, removing potential contributions from RON kinase activity, and recombinant EGFR kinase domain (EGFR-KD) as the active kinase. We found that EGFR-KD directly phosphorylated RON-K1114M, in an ATP-dependent and EGFR-KD concentration-dependent manner (*Figure 6C*).

## RON cannot substitute as activator or receiver in EGFR dimers

Structural studies have established the critical role for the orientation of EGFR kinase domains in asymmetric dimers (activator and receiver) for EGFR kinase activity (*Zhang et al., 2006*). We set out to determine if RON can substitute for either activator or receiver to form an active EGFR/RON heterodimer. HEK^RON cells were transfected with EGFR mutants that are either receiver-impaired (I682Q) or activator-impaired (V924R) (*Zhang et al., 2006*). For EGFR-WT, EGF stimulation resulted in the expected EGF-driven EGFR and RON phosphorylation patterns in HEK^RON cells (*Figure 7*). In contrast, neither EGFR-I682Q nor EGFR-V924R were capable of crosstalk with RON or EGFR autophosphorylation. As in previous studies (*Zhang et al., 2006*), restoring functional EGFR kinase domain dimers by co-expressing EGFR-I682Q and EGFR-V924R rescued EGFR autophosphorylation and – importantly – RON cross-phosphorylation. These data demonstrate that, unlike other ErbB family members that can form functional heterodimers with EGFR (*Kovacs et al., 2015b*), RON cannot serve as a substitute for the EGFR activator or receiver. Therefore, although EGFR can directly phosphorylate RON, this is not achieved through a simple hetero-dimerization event. Rather, these data indicate that the first step in crosstalk is for EGFR to form a signaling competent dimer in order to activate the EGFR kinase domain before phosphorylation of RON.

## Discussion

Our studies reveal that crosstalk between EGFR and RON occurs through direct receptor interaction, where EGFR transactivates RON within hetero-complexes. We also provide definitive evidence that crosstalk is EGF-driven and propagates in a unidirectional manner from EGFR to RON. Others have suggested that EGFR and RON can transactivate each other (*Hsu et al., 2006*; *Peace et al., 2003*).

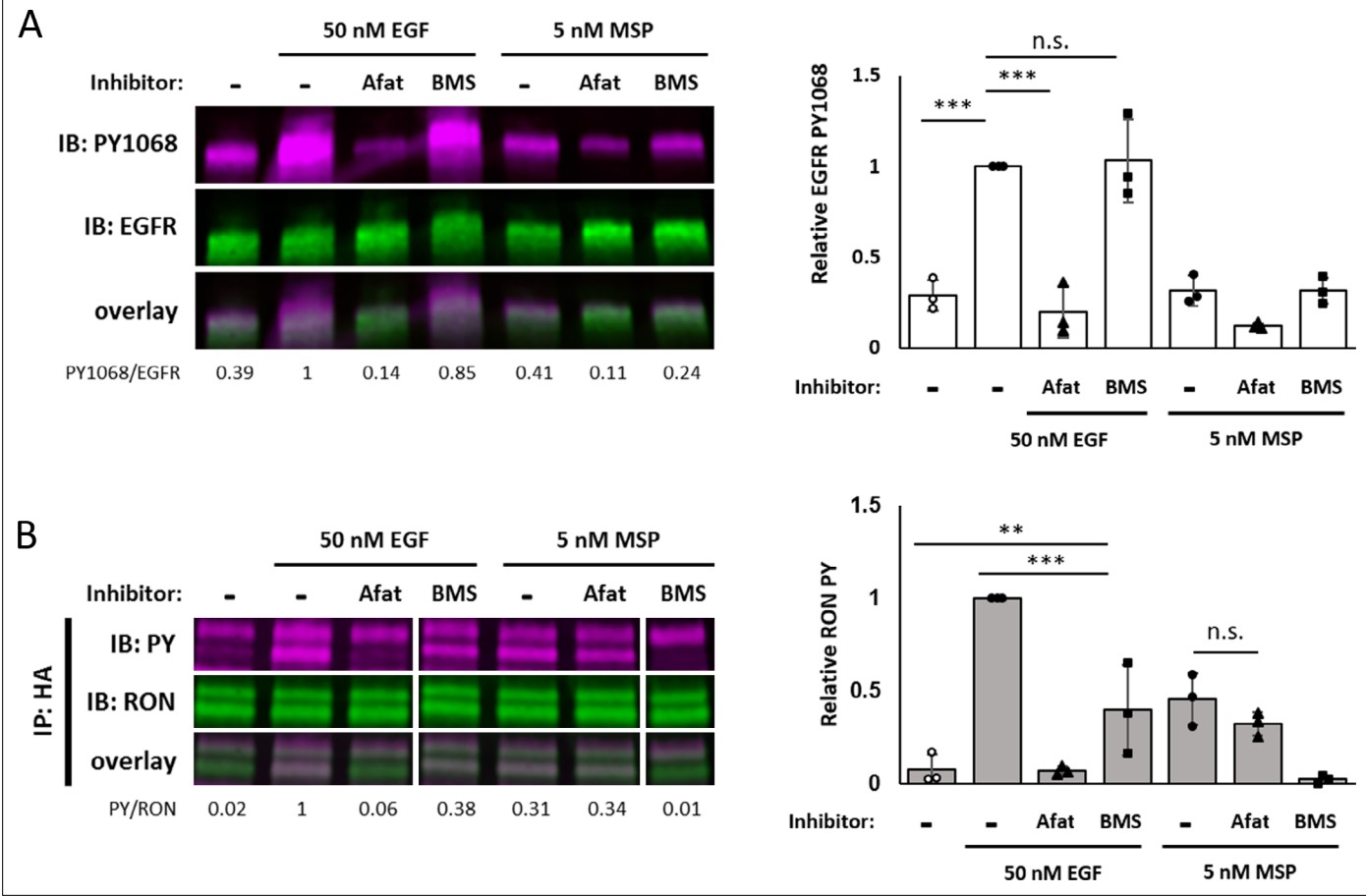

**Figure 5.** Maximal EGF-induced RON phosphorylation requires kinase activity of both receptors. (**A and B**) A431$^{RON}$ cells were pre-treated with 10 μM afatinib (Afat, pan-ErbB inhibitor) or 1 μM BMS777607 (BMS, Met family kinase inhibitor) for 20 or 15 min, respectively. Cells were then treated ± EGF or MSP for 5 min. (**A**) Cell lysates were used for PY1068 and EGFR immunoblots. (**B**) Lysates were immunoprecipitated (IP) with an anti-HA antibody and then immunoblotted for PY and RON. All samples are from the same blot, but an extraneous lane was removed for clarity. Bar graphs are corresponding mean ± SD from triplicate biological experiments. * p < 0.05; ** p < 0.01; *** p < 0.001.

The online version of this article includes the following source data and figure supplement(s) for figure 5:

**Source data 1.** Full raw western blots and blots with relevant bands labelled, corresponding to *Figure 5A,B*.

**Source data 2.** Source data for quantification of blots in *Figure 5A,B*.

**Figure supplement 1.** A431$^{RON-K1114M}$ (RON kinase-dead) cells were pre-treated, where indicated, with afatinib (Afat; pan-ErbB inhibitor) or BMS777607 (BMS; Met family kinase inhibitor) for 20 or 15 min, respectively.

**Figure supplement 1—source data 1.** Full raw western blots and blots with relevant bands labelled, corresponding to *Figure 5—figure supplement 1*.

**Figure supplement 1—source data 2.** Source data for quantification of blots in *Figure 5—figure supplement 1*.

**Figure supplement 2.** A431$^{RON}$ or A431$^{RON-K1114M}$ cells were treated ± EGF for 5 min before imaging.

One explanation for the previous findings could be cross-reactivity of the anti-phosphotyrosine antibodies used, since we found the commercially available phospho-RON 'receptor-specific' antibodies that we tested to be cross-reactive with phospho-RON and phospho-EGFR (see *Figure 1—figure supplement 4*). We avoided this potential artifact by ensuring that our protein analysis methods effectively resolved the contributions of RON separately from EGFR. We also considered the possibility that crosstalk could be dependent on the ratio of EGFR/RON levels, developing cell lines where EGFR is highly overexpressed compared to RON (~24:1) or where the expression is similar (~2:1). Notably, these model cell lines lack endogenous expression of other RON splice variants, allowing us to focus on interactions between wild type EGFR and wild type RON. In both cases, crosstalk was found to be

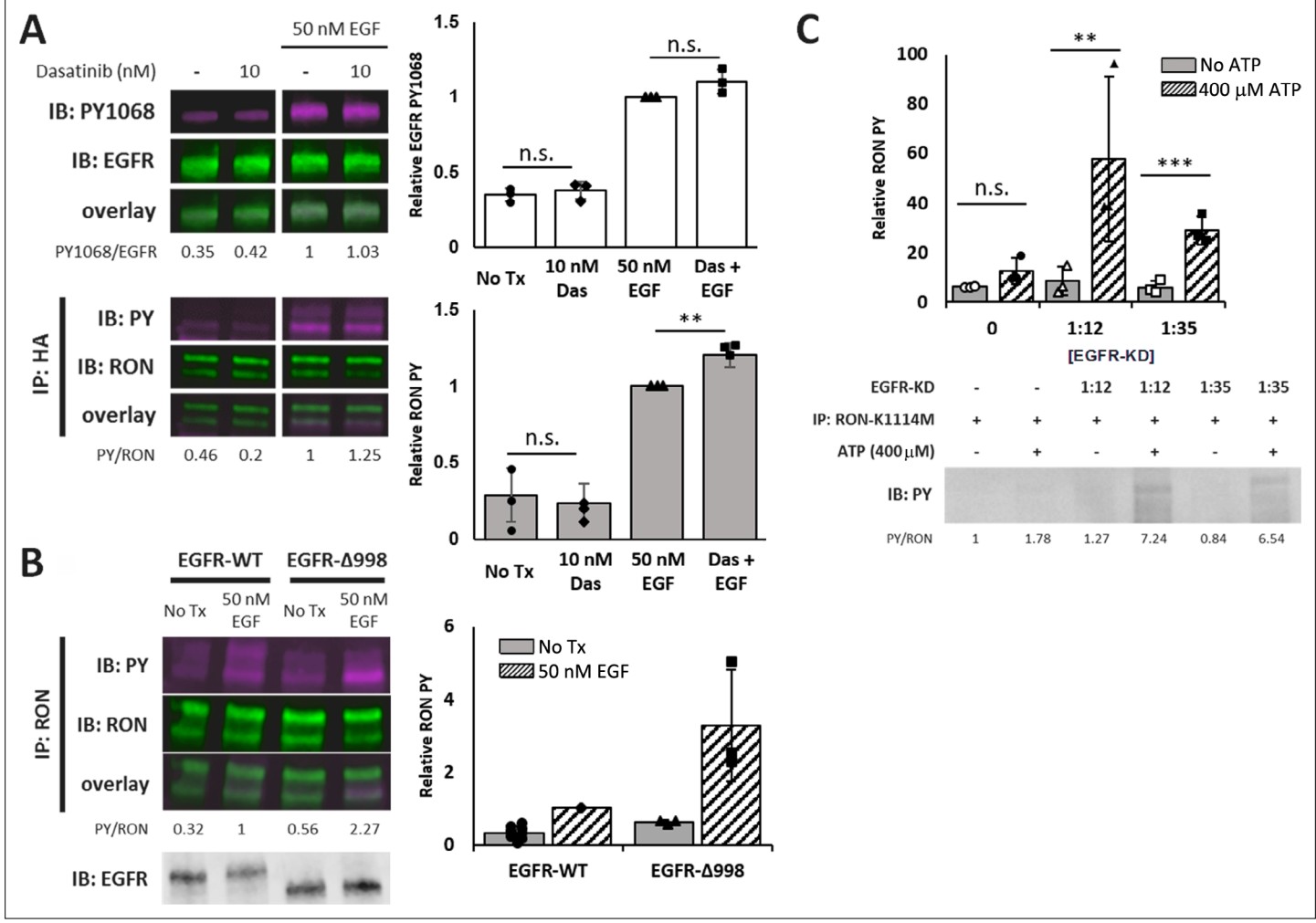

**Figure 6.** Crosstalk occurs through direct phosphorylation of RON by EGFR. (**A**) A431[RON] cells were pre-treated with dasatinib (Das, Src inhibitor) for 30 min prior to stimulation with EGF for 5 min at 37 °C. Representative immunoblots of cell lysates detecting PY1068 and total EGFR (top), or PY and RON after IP with anti-HA (RON) (bottom). (**B**) HEK[RON] cells transiently transfected with EGFR-WT or EGFR-Δ998 ± EGF for 5 min. Representative immunoblots detecting PY and RON after IP with anti-RON or detection of total EGFR on cell lysates (bottom inset). (**C**) Kinase assay using the purified EGFR kinase domain (EGFR-KD) co-incubated with RON-K1114M IP samples ± ATP. Representative immunoblot detecting total phosphorylation (PY) of RON. All bar graphs represent mean ± SD from triplicate biological experiments. ** p < 0.01; *** p < 0.001.

The online version of this article includes the following source data and figure supplement(s) for figure 6:

**Source data 1.** Full raw western blots and blots with relevant bands labelled, corresponding to *Figure 6A, B, and C*.

**Source data 2.** Source data for quantification of blots in *Figure 1C, D and E*.

**Figure supplement 1.** A431[RON] cells were pre-treated with dasatinib (Src inhibitor) at different concentrations for 30 min prior to cell lysis.

**Figure supplement 1—source data 1.** Full raw western blots and blots with relevant bands labelled, corresponding to *Figure 6—figure supplement 1*.

**Figure supplement 2.** Dephosphorylation assay was conducted using HEK[RON] cells transiently transfected with EGFR-WT or EGFR-Δ998.

**Figure supplement 2—source data 1.** Full raw western blots and blots with relevant bands labelled, corresponding to *Figure 6—figure supplement 2*.

unidirectional and EGF-dependent. Future studies are needed to define the role of crosstalk in situations where RON is more abundant than EGFR or different isoforms of RON are present.

An important outcome of our study is the first direct detection and quantification of the dynamic hetero-interactions between EGFR and RON. The use of two-color SPT allowed us to capture the formation and dissociation of EGFR/RON complexes on live cells and hetero-oligomerization was confirmed by correlated motion analysis. Other studies of EGFR/Met family crosstalk have inferred

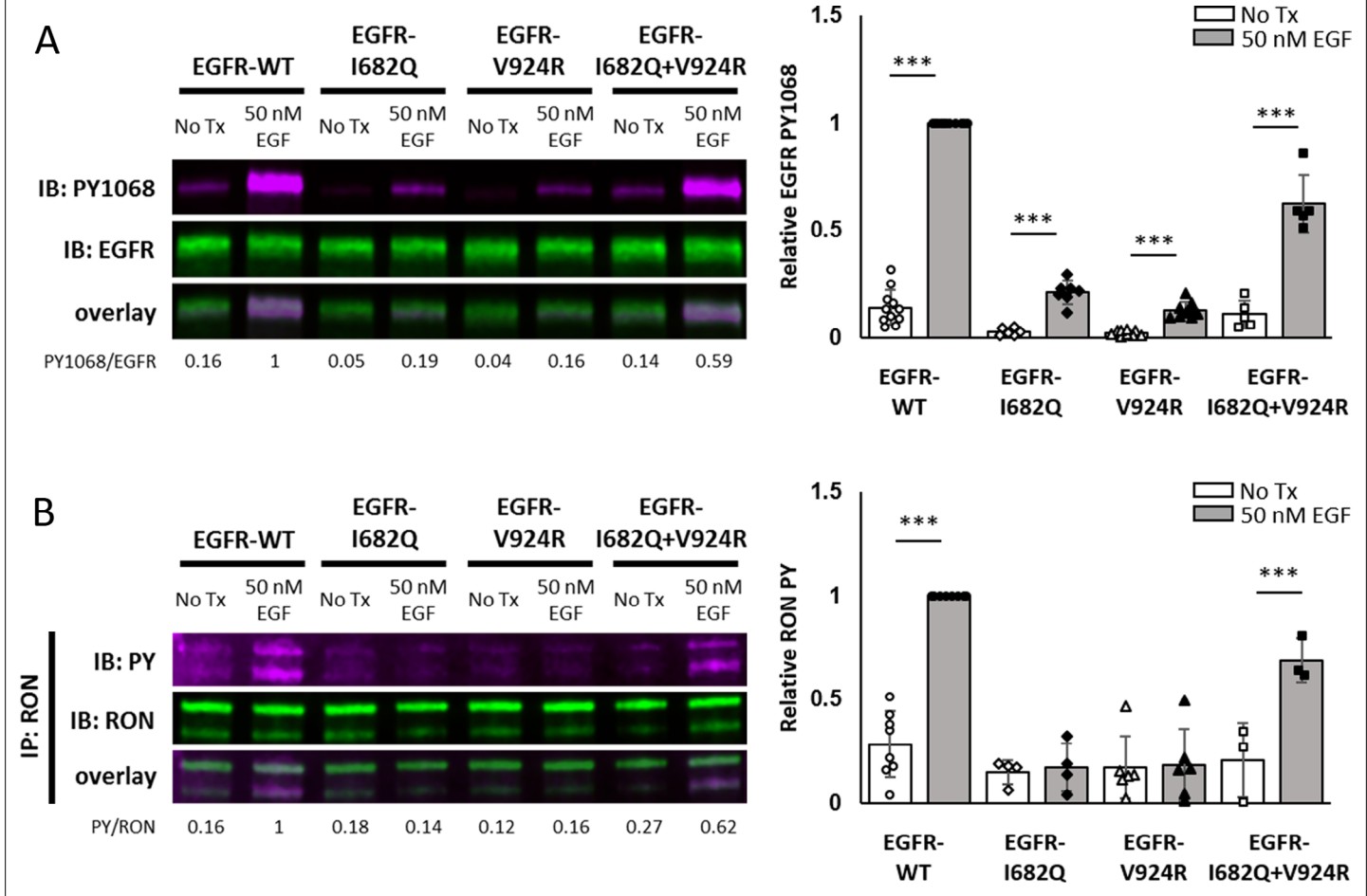

**Figure 7.** Functional EGFR dimers are necessary for EGFR/RON crosstalk. (**A and B**) HEK[RON] cells transiently expressing EGFR-WT, EGFR-I682Q (receiver-impaired), EGFR-V924R (activator-impaired) or both mutants (EGFR-I682Q + V924 R) were treated ± EGF for 5 min at 37 °C. (**A**) Representative immunoblot detecting PY1068 and EGFR in cell lysates. (**B**) Representative immunoblot showing PY and RON after IP with anti-RON. Triplicate biological experiments from (**A** and **B**) are quantified and graphed as mean ± SD. *** p < 0.001.

The online version of this article includes the following source data for figure 7:

**Source data 1.** Full raw western blots and blots with relevant bands labelled, corresponding to *Figure 7A, B*.

**Source data 2.** Source data for quantification of blots in *Figure 7*.

this interaction by co-IP or co-clustering in super-resolution imaging (*Harwardt et al., 2020*; *Jo et al., 2000*; *Peace et al., 2003*). Studies of EGFR/MET have also suggested that adaptor proteins downstream of the receptors, specifically c-Src, may mediate crosstalk (*Mueller et al., 2008*). It is also conceivable that adaptor proteins recruited to the EGFR tail (*Biscardi et al., 1999*; *Yamauchi et al., 1998*) could subsequently phosphorylate RON. However, we found that neither inhibition of c-Src activity nor removal of the EGFR cytoplasmic tail (EGFR-Δ998) prevented crosstalk with RON. Adaptor proteins may explain the enhanced phosphorylation of RON that was seen in cells expressing EGFR-Δ998. For instance, Grb2 has been reported to inhibit RON autophosphorylation (*Chaudhuri et al., 2011*) raising the possibility that loss of Grb2 recruitment by EGFR-Δ998 could reduce local Grb2 concentration and increase RON phosphorylation. Alternatively, removal of the EGFR C-terminal tail diminishes the recruitment of downstream EGFR substrates, limiting substrate competition and making the RON C-terminal tail the preferred substrate in the hetero-oligomeric complexes. Together, along with the identification of RON as a substrate for EGFR kinase, our results establish that crosstalk is mediated by receptor-receptor interactions. It is particularly intriguing that these interactions allow for EGFR to stimulate RON signaling in the absence of MSP and even when RON kinase activity is

inhibited. A potential future direction is to examine whether disruption of EGFR/RON interactions might provide a therapeutic advantage in tumors that co-express EGFR and RON.

The structural requirements for direct interactions between EGFR and RON are yet unresolved, but our studies have revealed important constraints governing these interactions. We found that RON cannot serve as an activator or receiver kinase in an EGFR/RON heterodimer. Instead, formation of a signaling-competent EGFR homodimer appears to be first required to initiate EGF-driven RON phosphorylation. Further study is needed to establish the exact stoichiometry and activity of the EGFR/RON complex. However, considering that RON homo-interactions were observed by two-color SPT in both resting and liganded states, we postulate that the hetero-complex consists of a RON dimer interacting with EGFR. Our studies with the EGFR dimer mutants suggest that the interaction involves either a ligand-bound EGFR dimer or an activated EGFR monomer that has recently dissociated from a homodimer.

Our findings suggest intriguing similarities between the interactions of EGFR with RON and those described for EGFR with ErbB3, a member of the EGFR subfamily. Studies from the Jura lab have proposed unidirectional receptor phosphorylation of unliganded ErbB3 by ligand-bound EGFR in which hetero-interactions are also thought to require EGFR dimers (*van Lengerich et al., 2017*). Furthermore, like EGFR and RON, EGFR and ErbB3 do not readily co-endocytose after EGF stimulation (*Lidke et al., 2004*). Therefore, the underlying mechanisms of EGFR/RON crosstalk are likely applicable to our understanding of other receptor interactions.

## Materials and methods

### Cell lines and reagents

Cell culture medium was from Thermo Fisher Scientific and Poly-L-lysine (PLL) from Sigma (cat # P4707). Afatinib and BMS777607 were from Selleck Chemicals (cat # S1011 and S1561, respectively), dasatinib from Santa Cruz Biotechnology (cat # sc-358114), and PD153035 from EMD Millipore (cat # 234491). Human recombinant EGF was from Invitrogen (cat # PHG0311) or PeproTech (cat # AF-100–15), biotin-conjugated and AF647-conjugated EGF from Thermo Fisher Scientific (cat # E3477 and E35351), and MSP from R&D Systems (cat # 4306 MS-010). Halt protease and phosphatase inhibitor (PPI) cocktail was from Pierce (cat # 78446) and the protease inhibitor cocktail set V, EDTA-free was from Calbiochem (cat # 539137). QD605 and QD655 streptavidin conjugates were from Thermo Fisher Scientific (cat # Q10101MP and Q10121MP, respectively). For western blotting, BCA protein assay kit (cat # 23225) and ECL blotting substrate (cat # 32106) were from Pierce. Immunoprecipitation was based on use of protein A/G magnetic beads from Pierce (cat # 88802). See Key Resources Table for a list of primary and secondary antibodies used in these studies.

Human epidermoid carcinoma A431 cells (ATCC, CRL-1555) were cultured in Dulbecco's Modified Eagle Medium (DMEM) supplemented with 10 % HyClone cosmic calf serum (CCS; GE Healthcare Life Sciences), 2 mM L-glutamine (Life Technologies), and penicillin/streptomycin (Life Technologies). Human embryonic kidney HEK-293 cells were cultured in Minimum Essential Medium (MEM) with 10 % fetal bovine serum (FBS; Atlanta Biologicals), 2 mM L-glutamine, and penicillin/streptomycin. Cell lines were authenticated using STR profiling (ATCC) and free from mycoplasma (MycoAlert Mycoplasma Detection Kit; Lonza).

### Plasmid cloning, site directed mutagenesis and cell transfections

The vector containing RON (MST1R) pDONR223-MST1R was a gift from William Hahn and David Root (Addgene plasmid # 23942; http://n2t.net/addgene:23942; RRID:Addgene_23942) (*Johannessen et al., 2010*). HA-tagged RON was cloned into the expression vector pcDNA3.1/V5-His-TOPO (Invitrogen) by fusion PCR. An ultramer containing the CACC ligation sequence, start codon, RON signal peptide, HA-tag, and alanine linker 5' of the mature RON coding region and a reverse primer were used to synthesize HA-RON. DNA oligos were from Integrated DNA Technologies. Ultramer sequencing and mutagenesis primers are listed in **Key Resources Table**. The kinase dead RON variant (HA-RON-K1114M) was generated by site-directed mutagenesis (*Danilkovitch-Miagkova et al., 2000*) (**Key Resources Table**). To establish cell lines stably expressing HA-RON (HEK$^{RON}$ and A431$^{RON}$), cells were transfected with the pcDNA3.1 HA-RON plasmid by electroporation using the AMAXA Nucleofector System (Lonza). Briefly, $5 \times 10^6$ HEK-293 cells were transfected with 8 μg of plasmid

DNA using Nucleofection Solution V and program Q-001. A431 cells were transfected with HA-RON or HA-RON-K1114M using solution T and program X-001. Transfected cells were selected for stable integration by growth in 1 mg/ml G418 (Caisson Labs) for 7 days, then sorted for RON expression with a fluorescently-conjugated anti-HA antibody using a iCyt SY3200 cell sorter (Sony Biotechnology).

For co-expression of RON and EGFR, HEK^RON cells were transfected with an ACP-tagged EGFR plasmid (*Valley et al., 2015*) by electroporation using the same conditions as above. Transfected cells were selected with zeocin (300 μg/ml; Gibco/Life Technologies) and sorted for double positive cells (anti-HA-AF488 and anti-EGFR-AF647) on the iCyt SY3200.

For kinase assays, a C-terminal SBP-tagged construct of EGFR encoding the transmembrane domain, kinase domain, and cytoplasmic tail (EGFR-KD) was amplified from full-length EGFR by PCR (**Key Resources Table**) and cloned into the pCTAP backbone via Gibson assembly. EGFR-KD and RON-K1114M proteins were produced using the Expi293 cell Expression System (Thermo Fisher Scientific) according to the manufacturer's recommendations.

Receiver-impaired and activator-impaired EGFR variants, EGFR-I682Q and EGFR-V924R, were engineered from the pcDNA3.1 HA-EGFR WT plasmid using site-directed mutagenesis (*Valley et al., 2015*) (**Key Resources Table**). The truncated EGFR-Δ998 plasmid, which lacks the C-terminal phosphorylation sites, was generated by amplifying the truncated EGFR from pcDNA3.1-EGFR WT plasmid using standard PCR and cloning techniques (**Key Resources Table**). HEK^RON cells were transiently transfected with the resulting plasmids and experiments performed at 18–24 hr post-transfection.

## Flow cytometry – receptor quantification

Quantification of cell surface EGFR and RON expression was performed by flow cytometry using Quantum MESF kits. Briefly, cells were incubated with a range of concentrations (0–40 μg/ml) of anti-EGFR-AF647 (dye/protein ratio of 2.74 or 3.84) or anti-HA-AF488 (dye/protein ratio of 3.34) for 1 hr on ice. Cells were rinsed with PBS, fixed in 4 % PFA (paraformaldehyde) for 10 min on ice, washed with 10 mM Tris-PBS and resuspended in PBS. Fluorescent calibrator beads, Quantum AlexaFluor 647 or 488 MESF (Bangs Laboratories, cat # 647 A and 488 A, respectively) were used to generate a standard curve of fluorescence intensity. Samples and beads were run on the Accuri C6 Plus cytometer (BD Biosciences), and receptor levels calculated based on the dye:protein ratio of the individual antibodies and values determined using the QuickCal spreadsheet (Bangs Laboratories).

## Immunofluorescence staining

HEK^RON/EGFR cells were plated onto glow-discharged (EMS 150T ES, Quorum Technologies), PLL-coated glass coverslips overnight. RON labeling was performed in live cells with an anti-HA-FITC Fab fragment for 30 min in Tyrodes buffer (135 mM NaCl, 10 mM KCl, 0.4 mM MgCl$_2$, 1 mM CaCl$_2$, 10 mM HEPES, 20 mM glucose, 0.1% BSA, pH 7.2) on ice. Cells were treated with 10 nM EGF-AF647 on ice for 5 min, fixed in 4% PFA for 15 min at RT, and washed with 10 mM Tris/PBS buffer. Samples were rinsed, incubated with DAPI, and mounted with Prolong Gold (Thermo Fisher Scientific). Confocal images were acquired using a 63×/1.40 oil objective on a Zeiss LSM800 microscope in channel mode and appropriate diode lasers were used for excitation of the fluorophores.

For endocytosis experiments, RON was pre-labeled with anti-HA-FITC Fab and cells were stimulated with EGF-AF647 for 10 min at 37 °C prior to fixation. Samples were simultaneously blocked and permeabilized with 0.1% Triton X-100/3% BSA/PBS for 20 min and stained with anti-EEA1 in 0.1% Triton X-100/0.1% BSA/PBS solution for 30 min at 37 °C followed by anti-Rabbit-AF555 secondary for 30 min at 37 °C before DAPI staining and mounting.

## Transmission electron microscopy of native membrane sheets

Standard 'rip-flip' membrane sheets were prepared as previously described (*Wilson et al., 2007*). In brief, A431^RON cells were treated or not with 50 nM EGF for 2 or 5 min and fixed in 0.5% PFA. Coverslips were flipped, cells down, onto PLL-coated formvar and carbon-coated nickel finder grids and pressure was applied to adhere apical cell membranes before removing the coverslip. Grids with membrane sheets were fixed with 2% PFA in HEPES buffer (25 mM HEPES, 25 mM KCl, and 2.5 mM Mg Acetate) for 20 min and sequentially labeled with antibodies against RON or EGFR in 0.1% BSA/PBS for 1 h at RT. Secondary antibodies conjugated to colloidal gold were added for 30 min at RT. Samples were post-fixed with 2% glutaraldehyde for 20 min and negatively stained with 0.3% tannic acid for 1 min

and 2% uranyl acetate for 9 min. Digital images were acquired on a Hitachi H-7650 Transmission Electron Microscope equipped with a mid-mount digital imaging system (Advanced Microscopy Techniques, Corp) and Image J (NIH) was used to crop images. Ripley's bivariate K test was used to determine if co-clustering of species is significant (*Wilson et al., 2004*; *Yang et al., 2007*), with a critical interaction distance of 50 nm. Data within the confidence window are not significantly co-clustered. When the experimental values are found above the confidence window the deviation from complete spatial randomness is statistically significant and indicates that the two labels are co-clustering.

## Cell activation and lysis

Transiently transfected or stable cell lines were seeded into 100 mm dishes and allowed to adhere overnight. For inhibition studies, cells were pretreated with 10 µM afatinib for 20 min, 1 µM BMS777607 for 15 min, or 1–10 nM dasatinib for 30 min, where indicated. They were subsequently treated with different doses of EGF, MSP, or both, for varying times (0–5 min). Cells were rinsed in cold PBS and lysed on ice for 20 min with NP-40 lysis buffer (150 nM NaCl, 50 mM Tris, 1% NP-40) containing PPI. Lysates were cleared and protein concentrations in the supernatant were determined by BCA protein assay.

## Immunoprecipitation

Cell lysates (1 mg total protein) were immunoprecipitated (IP) overnight using anti-HA coupled to magnetic or sepharose beads or anti-RON overnight at 4 °C, rotating. For samples incubated with the RON antibody, protein A/G magnetic beads were added the next day and incubated for 1 h, rotating at 4 °C. Beads were washed with 0.05% Tween-20/ PBS containing PPI.

## Multiplex immunoblotting

Whole lysates (20 µg) or IP samples were boiled with reducing sample buffer, subject to SDS-PAGE, and transferred to nitrocellulose membranes using the iBlot2 system (Life Technologies). Membranes were blocked for 30 min in 3% BSA / 0.1% Tween-20/ TBS, and probed overnight with primary antibodies at 4 °C (Key Resources Table). Membranes were incubated with IRDye fluorescent secondary antibodies for 1 h at RT (Key Resources Table), washed, and dual color detection was performed using the Odyssey Fc Imaging System (Li-Cor). Band intensities were analyzed with Image Studio (Li-Cor, version 5.2) and normalized PY to total protein (PY1068/EGFR or PY/RON).

## Single particle tracking (SPT)

Single- and dual-color SPT and analysis was conducted as previously described (*Low-Nam et al., 2011*; *Steinkamp et al., 2014*; *Valley et al., 2015*). Briefly, A431$^{RON}$ (Figures 3A, B , and 4) or A431$^{RON-K1114M}$ (*Figure 5—figure supplement 2*) cells were seeded in eight-well chamber slides (Nunc Lab-Tek) at a density of 30,000/well and allowed to adhere overnight. Where indicated, EGFR kinase activity was inhibited by pretreating with 1 µM PD153035 for 2 hr and maintained throughout the experiment. RON was tracked via QD conjugated to biotinylated anti-HA Fab fragments that bind to the N-terminal HA-tag on HA-RON (as indicated). Cells were incubated with 200 pM anti-HA-QDs (605 or 655) for 15 min at 37 °C to obtain single-molecule density on the apical surface. After washing with Tyrodes buffer cells were treated with 5 nM MSP for 5 min or 50 nM EGF for 30 s and imaged. For dual EGFR and RON tracking, cells were incubated with 200 pM anti-HA-QD655 for 15 min at 37 °C, washed, and then stimulated with 50 pM QD605-conjugated EGF-biotin. Particle tracking was done for up to 15 min. Imaging was performed on an Olympus IX71 inverted widefield microscope with a 60× 1.2 numerical aperture water objective as in *Valley et al., 2015* (*Valley et al., 2015*). QD emissions were collected using a 600 nm dichroic (Chroma) and the appropriate bandpass filters, 600/52 nm and 676/37 (Semrock). Physiological temperature (34–36°C) was maintained using an objective heater (Bioptechs). Images were acquired at a rate of 20 frames per sec for a total movie length of 1000 frames.

MATLAB (MathWorks) was used for image processing and analysis in conjunction with DIPImage (Delft University of Technology). Diffusion was computed using mean square displacement (MSD) (*de Keijzer et al., 2008*; *Low-Nam et al., 2011*; *Valley et al., 2015*). Dimer off-rates and events were identified using a two-state HMM, similar to previous work (*Low-Nam et al., 2011*; *Steinkamp et al., 2014*). For more details, see Supplementary Methods.

## Protein purification and kinase assay

EGFR-KD, which begins at amino acid 637 and continues through the C-terminal tail, was expressed in Expi293 cells. Cell lysate was bound to streptavidin resin and eluted in biotin buffer according to manufacturer's recommendations (InterPlay Mammalian TAP System; Agilent Technologies). Typical protein yield was between 30–70 µg. RON-K1114M was immunoprecipitated from Expi293 cell lysates (2 mg total protein) with sepharose anti-HA beads. Immunoprecipitated RON-K1114M was resuspended in kinase assay buffer (200 mM HEPES, pH 7.4; 300 mM $MgCl_2$; 20 mM $MnCl_2$; 0.5 % Triton X-100; 1.5 % Brij 35; 10 % glycerol; 1 X Protease inhibitor cocktail Set V; and 2 mM activated $Na_3VO_4$) in the presence or absence of purified EGFR-KD (1:12 or 1:35 dilution). Samples were incubated with 400 µM ATP (or no ATP, as a control; Cell Signaling Technology, cat # 9804) and held at 30 °C for 30 min, shaking. Reactions were terminated by addition of ice cold buffer. RON-K1114M bound to beads was recovered by centrifugation at 2500 x g for 2 min at 4 °C and washed 3 x with 0.05 % Tween-20/ PBS containing PPI. Samples were boiled with reducing sample buffer, subject to SDS-PAGE, and western blotting with HRP-conjugated anti-PY20 and anti-PY99.

## Mass spectrometry

A431[RON] cells were harvested with lysis buffer and immunoprecipitated with an anti-HA antibody. Samples were run in a reducing 4–20% polyacrylamide gel for separation, washed in distilled water for 15 min, and incubated with GelCode Blue stain reagent (Thermo Fisher Scientific, cat # 24590) for 1 hr at RT. Both top and bottom RON bands were excised from the gel and samples sent to the Proteomics Core at UT Southwestern.

## Dephosphorylation assay

HEK[RON] cells were transiently transfected with WT or Δ998-EGFR and allowed to attach and recover overnight. Cells were activated with 50 nM EGF for 2 min followed by 10 µM afatinib for 20 or 40 s. Media was removed and reactions were stopped by placing plates on top of a layer of liquid nitrogen. Protein lysates were harvested and quantified by BCA. RON protein was immunoprecipitated from the lysates with anti-RON antibody, and immunoblotted.

## Statistical analysis

All values from quantitative western blot experiments are plotted as mean ± SD. For quantitative experiments, statistical analysis was performed using GraphPad Prism (Prism 4, GraphPad) with a two-way analysis of variance (ANOVA) from three biological replicates (performed on separate days). For immunoblot analysis, phosphorylated protein levels were normalized to total protein levels (RON or EGFR) detected from the same sample. For phosphorylation time course experiments, the maximum stimulation level was set at one for triplicate experiments and plotted as mean ± SD. Differences among means were tested using the Bonferroni multiple comparison test post hoc. Values of $p < 0.05$ were considered significant. Errors in values of diffusion coefficients are reported as 95 % confidence intervals from fitting a Brownian diffusion model (linear) to the first 5 points of the MSD.

# Acknowledgements

We thank Dr. Peter Cooke of the Electron Microscopy Laboratory at New Mexico State University for access to a Hitachi H-7650 TEM. Mass spectrometry data was generated at the UT Southwestern Proteomics Core. We thank Shayna Lucero for assistance with cell culture and flow cytometry, Dr. Michael Wester for assistance with EM analysis, Dr. Mohamad Fazel for assistance with single particle tracking analysis, Eric Burns for assistance with SPT data collection, Danielle Burke for western blot assistance, and Dr. Chris Valley and Russell Hunter for plasmid preparations. We thank Dr. Mark Lemmon for helpful discussions and suggestions on the manuscript. We gratefully acknowledge use of the University of New Mexico Comprehensive Cancer Center fluorescence microscopy and flow cytometry facilities, as well as the NIH P30CA118100 support for these cores.

## Additional information

### Funding

| Funder | Grant reference number | Author |
|---|---|---|
| National Institutes of Health | R35GM126934 | Diane S Lidke |
| National Institutes of Health | R21GM132716 | Keith Lidke |
| New Mexico Spatiotemporal Modeling Center | P50GM085273 | Bridget S Wilson Diane S Lidke |
| University of New Mexico | Undergraduate Pipeline Network P30CA118100 | Justine M Keth Aubrey C Gibson |
| ASERT-IRACDA | K12GM088021 | Elton D Jhamba |
| National Institutes of Health | R01CA248166 | Diane S Lidke |
| University of New Mexico | MARC Program 2T34GM008751 | Justine M Keth |

The funders had no role in study design, data collection and interpretation, or the decision to submit the work for publication.

### Author contributions

Carolina Franco Nitta, Conceptualization, Data curation, Formal analysis, Investigation, Methodology, Resources, Writing – original draft, Writing – review and editing; Ellen W Green, Conceptualization, Investigation, Methodology, Resources, Writing – review and editing; Elton D Jhamba, Rachel M Grattan, Formal analysis, Investigation, Methodology, Writing – review and editing; Justine M Keth, Iraís Ortiz-Caraveo, Aubrey C Gibson, Investigation, Methodology, Writing – review and editing; David J Schodt, Data curation, Formal analysis, Software, Writing – review and editing; Ashwani Rajput, Conceptualization, Writing – review and editing; Keith A Lidke, Data curation, Formal analysis, Software, Supervision, Writing – review and editing; Bridget S Wilson, Conceptualization, Funding acquisition, Supervision, Writing – original draft, Writing – review and editing; Mara P Steinkamp, Conceptualization, Data curation, Formal analysis, Resources, Supervision, Writing – original draft, Writing – review and editing; Diane S Lidke, Conceptualization, Data curation, Formal analysis, Funding acquisition, Project administration, Supervision, Writing – original draft, Writing – review and editing

### Author ORCIDs

Carolina Franco Nitta (iD) http://orcid.org/0000-0002-9122-5453
Ellen W Green (iD) http://orcid.org/0000-0002-2083-522X
David J Schodt (iD) http://orcid.org/0000-0002-8986-2736
Mara P Steinkamp (iD) http://orcid.org/0000-0003-1226-9325
Diane S Lidke (iD) http://orcid.org/0000-0001-8533-6029

### Decision letter and Author response

Decision letter https://doi.org/10.7554/eLife.63678.sa1
Author response https://doi.org/10.7554/eLife.63678.sa2

## Additional files

### Supplementary files

• Transparent reporting form

• Supplementary file 1. List of top proteins co-IP in A431RON cells with anti-HA antibody for RON pulldown, as analyzed by Mass Spectrometry.

## Data availability

All data generated or analyzed during this study are included in the manuscript and supporting files. Source data for the quantitative plots and gels have been provided.

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

# Appendix 1

## Appendix 1—key resources table

| Reagent type (species) or resource | Designation | Source or reference | Identifiers | Additional information |
|---|---|---|---|---|
| cell line (*Homo sapiens*) | A431 (female) | ATCC | ATCC Cat# CRL-1555, RRID:CVCL_0037 | |
| cell line (*Homo sapiens*) | HEK-293 | ATCC | ATCC Cat# CRL-1573, RRID:CVCL_0045 | |
| cell line (*Homo sapiens*) | A431<sup>RON</sup> | This paper | | A431 cells stably transfected with pcDNA3.1 HA-RON plasmid |
| cell line (*Homo sapiens*) | A431<sup>RON-K1114M</sup> | This paper | | A431 cells stably transfected with pcDNA3.1 HA-RON-K1114M plasmid |
| cell line (*Homo sapiens*) | HEK<sup>RON</sup> | This paper | | HEK-293 cells stably transfected with pcDNA3.1 HA-RON plasmid |
| cell line (*Homo sapiens*) | HEK<sup>RON/EGFR</sup> | This paper | | HEK<sup>RON</sup> cells stably transfected with ACP-EGFR plasmid |
| transfected construct (human) | pcDNA3.1 HA-RON | This paper | | Generated using pcDNA3.1 |
| transfected construct (human) | ACP-EGFR | Ziomkiewicz et al., Cytometry A, 2013 | | |
| transfected construct (human) | pcDNA3.1 HA-RON-K1114M | This paper | | |
| antibody | Anti-EEA1 (Rabbit monoclonal) | Cell Signaling Technology | Cat# 3288 RRID:AB_2096811 | Clone C5B10 IF (1:200) |
| antibody | Anti-EGFR (Rabbit monoclonal) | Cell Signaling Technology | Cat# 4267 RRID:AB_2246311 | Clone D38B1 WB (1:2000) |
| antibody | Anti-EGFR (Goat polyclonal) | R&D Systems | Cat# AF231 RRID:AB_355220 | WB (1:1000) Used when blotting for EGFR-Δ998 |
| antibody | Anti-EGFR (Rabbit monoclonal) | Cell Signaling Technology | Cat# 4405 RRID:AB_331380 | Clone 15 F8 WB (1:2000) |
| antibody | Anti-EGFR PY1068 (Mouse monoclonal) | Cell Signaling Technology | Cat# 2236 RRID:AB_331792 | Clone 1 H12 WB (1:2000) |
| antibody | Anti-EGFR PY845 (Rabbit polyclonal) | Santa Cruz Biotechnology | Cat# sc-23420 RRID:AB_653168 | WB (1:500) |
| antibody | Anti-EGFR PY1148 (Rabbit polyclonal) | Cell Signaling Technology | Cat# 4404 RRID:AB_331127 | WB (1:2000) |
| antibody | Anti-EGFR AF647 (Mouse monoclonal) | Santa Cruz Biotechnology | Cat# sc-101 AF647 | Clone R-1 FACS (5–40 ug/mL) |
| antibody | Anti-EGFR (Goat polyclonal) | Santa Cruz Biotechnology | Cat# sc-31156 RRID:AB_2096710 | Clone D-20 EM (1:20) |

*Appendix 1 Continued on next page*

*Appendix 1 Continued*

| Reagent type (species) or resource | Designation | Source or reference | Identifiers | Additional information |
|---|---|---|---|---|
| antibody | Anti-HA AF488 (Mouse Monoclonal) | Cell Signaling Technology | Cat# 2350 RRID:AB_491023 | Clone 6E2 FACS (5–40 ug/mL) |
| antibody | Anti-HA magnetic bead (Rabbit monoclonal) | Cell Signaling Technology | Cat# 11846 RRID:AB_2665471 | Clone C29F4 IP (1:100) |
| antibody | Anti-HA sepharose bead (Rabbit monoclonal) | Cell Signaling Technology | Cat# 3956 RRID:AB_10695091 | Clone C29F4 IP (1:100) |
| antibody | Anti-HA FITC (Fab; Rat monoclonal) | Roche | Cat# 11988506001 RRID:AB_390916 | Clone 3 F10 IF (1:20) |
| antibody | Anti-HA Biotin (Fab; Rat monoclonal) | Roche | Cat# 12158167001 RRID:AB_390915 | Clone 3 F10 SPT (200 pM) |
| antibody | Anti-PY20 (Mouse monoclonal) | Santa Cruz Biotechnology | Cat# sc-508 RRID:AB_628122 | WB (1:500) |
| antibody | Anti-PY20 HRP (Mouse monoclonal) | Santa Cruz Biotechnology | Cat# sc-508 HRP | Kinase assay (1:500) |
| antibody | Anti-PY99 (Mouse monoclonal) | Santa Cruz Biotechnology | Cat# sc-7020 RRID:AB_628123 | WB (1:500) |
| antibody | Anti-PY99 HRP (Mouse monoclonal) | Santa Cruz Biotechnology | Cat# sc-7020 HRP | Kinase assay (1:500) |
| antibody | Anti-RON (Goat polyclonal) | R&D Systems | Cat# AF691 RRID:AB_355527 | IP (1:100) |
| antibody | Anti-RONβ (Rabbit polyclonal) | Santa Cruz Biotechnology | Cat# sc-322 RRID:AB_677390 | Clone C-20 WB (1:500) EM (1:20) Discontinued antibody; remainder of experiments done with Cell Signaling Technology Cat# 2,654 |
| antibody | Anti- RONβ (Rabbit monoclonal) | Cell Signaling Technology | Cat# 2654 RRID:AB_2298153 | Clone C81H9 WB (1:2000) |
| antibody | Anti- RONβ (Mouse monoclonal) | Santa Cruz Biotechnology | Cat# sc-74588 RRID:AB_2235711 | Clone E-3 WB (1:500) Used with the PY1238/39 RON antibody (R&D Systems Cat# AF1947) |
| antibody | Anti-RON PY1238/39 (Rabbit polyclonal) | R&D Systems | Cat# AF1947 RRID:AB_1152159 | WB (1:2000) |
| antibody | Anti-Src (Mouse monoclonal) | Cell Signaling Technology | Cat# 2110 RRID:AB_10691385 | Clone L4A1 WB (1:2000) |
| antibody | Anti-Src PY416 (Rabbit polyclonal) | Cell Signaling Technology | Cat# 2101 RRID:AB_331697 | WB (1:2000) |
| antibody | Anti-Goat IgG IRDye 800CW (Donkey polyclonal) | Li-Cor | Cat# 926–32214 RRID:AB_621846 | WB (1:20000) |

*Appendix 1 Continued on next page*

*Appendix 1 Continued*

| Reagent type (species) or resource | Designation | Source or reference | Identifiers | Additional information |
|---|---|---|---|---|
| antibody | Anti-Goat IgG 12 nm colloidal gold (Donkey polyclonal) | Jackson Immuno Research | Cat# 705-205-147 RRID:AB_2340418 | EM (1:20) |
| antibody | Anti-Mouse IgG IRDye 680RD (Goat polyclonal) | Li-Cor | Cat# 926–68070 RRID:AB_10956588 | WB (1:20000) |
| antibody | Anti-Mouse IgG IRDye 680RD (Donkey polyclonal) | Li-Cor | Cat# 926–68072 RRID:AB_10953628 | WB (1:20000) |
| antibody | Anti-Rabbit IgG IRDye 800CW (Goat polyclonal) | Li-Cor | Cat# 926–32211 RRID:AB_621843 | WB (1:20000) |
| antibody | Anti-Rabbit IgG IRDye 680RD (Donkey polyclonal) | Li-Cor | Cat# 926–68073 RRID:AB_10954442 | WB (1:20000) |
| antibody | Anti-Rabbit IgG AF555 (Fab; Goat polyclonal) | Thermo Fisher Scientific | Cat# A-21430 RRID:AB_2535851 | IF (1:500) |
| antibody | Anti-Rabbit IgG 6 nm colloidal gold (Donkey polyclonal) | Jackson Immuno Research | Cat# 711-195-152 RRID:AB_2340609 | EM (1:20) |
| recombinant DNA reagent | pDONR223-MST1R | Addgene | RRID:Addgene_23942 *Johannessen et al., 2010* | RON sequence used to make the HA-tagged RON plasmid |
| recombinant DNA reagent | pcDNA3.1/V5-His-TOPO | Invitrogen | | Used as a backbone for HA-tagged RON plasmid (HA-RON and HA-RON-K1114M) |
| recombinant DNA reagent | pcTAP | Agilent | | |
| recombinant DNA reagent | pcDNA3.1 HA-EGFR WT | *Valley et al., 2015* | | Plasmid used for generating EGFR-I682Q, EGFR-V924R, and EGFR-Δ998 with primers below |
| sequence-based reagent | Ultramer to generate HA-tagged RON plasmid | This paper | | CACCATGGAGCTCCTC CCGCCTCAGTCCTTCC TGTTGCTGCTGCTGTT GCCTGACAAGCCCGCG GCGGGCTATCCTTACG ACGTGCCTGACTACGCC GCAGCAGCAGAGGACT GGCAGTGCCCGCACA Has CACC ligation sequence, start codon, RON signal peptide, HA-tag, and alanine linker 5' of the mature RON coding region |
| sequence-based reagent | Primers to generate RON-K1114M mutagenesis | This paper | *Danilkovitch-Miagkova et al., 2000* | Forward: GTGATGCGAC TTAGTGACATGATGGC ACATTGGATTC Reverse: GAATCCAATG TGCCATCATGTCACTAA GTCGCATCAC |

*Appendix 1 Continued on next page*

*Appendix 1 Continued*

| Reagent type (species) or resource | Designation | Source or reference | Identifiers | Additional information |
|---|---|---|---|---|
| sequence-based reagent | Primers to generate EGFR-KD | This paper | | Forward: CGCCGGATCC CCAACGAATGGGCCTA AG Reverse: CGAGGTCGAC GGTATCGATAAGCTTTG CTCCAATAAATTCACTGC |
| sequence-based reagent | Primers to generate EGFR-I682Q mutagenesis | This paper | | Forward: CAACCAAGCT CTCTTGAGGCAGTTG AAGGAAACTGAATTC Reverse: GAATTCAGT TTCCTTCAACTGCCTC AAGAGAGCTTGGTTGG |
| sequence-based reagent | Primers to generate EGFR-V924R mutagenesis | This paper | | Forward: GATGTCTACA TGATCATGCGCAAGT GCTGGATGATA Reverse: TATCATCCAG CACTTGCGCATGATC ATGTAGACATC |
| sequence-based reagent | Primers to generate truncated EGFR-Δ998 | This paper | | Forward: GTTAAGCTTG GTACCGAGCTCGGAT CCAGTACCCTTCACC ATGCGACCCTCCGGG AC Reverse: CCCTCTAGA CTCGAGCGGCCGCCT AGAAGTTGGAGT CTGTAGGACTTGGC |
| peptide, recombinant protein | Human recombinant EGF | Invitrogen | Cat# PHG0311 | |
| peptide, recombinant protein | Human recombinant EGF | PeproTech | Cat# AF-100–15 | |
| peptide, recombinant protein | Human recombinant EGF-biotin | Thermo Fisher Scientific | Cat# E3477 | |
| peptide, recombinant protein | Human recombinant EGF-AF647 | Thermo Fisher Scientific | Cat# E35351 | |
| peptide, recombinant protein | Human recombinant MSP | R&D Systems | Cat# 4306 MS-010 | |
| commercial assay or kit | BCA protein assay kit | Pierce | Cat# 23,225 | |
| commercial assay or kit | ECL blotting substrate | Pierce | Cat# 32,106 | |
| commercial assay or kit | Expi293 Expression System Kit | Thermo Fisher Scientific | Cat# A14635 | Used for producing EGFR-KD and RON-K1114M proteins for use in kinase assay |
| chemical compound, drug | Afatinib | Selleck Chemicals | Cat# S1011 | |
| chemical compound, drug | BMS777607 | Selleck Chemicals | Cat# S1561 | |
| chemical compound, drug | Dasatinib | Santa Cruz Biotechnology | Cat# sc-358114 | |
| chemical compound, drug | PD153035 | EMD Millipore | Cat# 234,491 | |
| software, algorithm | MATLAB | Mathworks | RRID:SCR_001622 | |
| software, algorithm | DIPImage | Delf University of Technology | | |

*Appendix 1 Continued on next page*

*Appendix 1 Continued*

| Reagent type (species) or resource | Designation | Source or reference | Identifiers | Additional information |
|---|---|---|---|---|
| other | QD605 streptavidin | Thermo Fisher Scientific | Cat# Q10101MP | For use in SPT |
| other | QD655 streptavidin | Thermo Fisher Scientific | Cat# Q10121MP | For use in SPT |
| other | protein A/G magnetic beads | Pierce | Cat# 88,802 | For use in IP |
| other | Quantum AlexaFluor 647 | Bangs Laboratories | Cat# 647 A | For use in receptor quantification on FACS |
| other | Quantum AlexaFluor 488 | Bangs Laboratories | Cat# 488 A | For use in receptor quantification on FACS |

