## [Editor Report]

The study by Nitta et al., brings a sophisticated understanding of the mechanisms behind crosstalk between two mitogenic growth factor receptor tyrosine kinases – EGFR and RON. While earlier studies indicated that the receptors interact, this work provides evidence that the EGFR-RON crosstalk is unidirectional. They also show that this interaction takes place specifically at the membrane, rule out other intermediary molecules and locations between EGFR and RON. Consistent with this they find that, in vitro, RON acts as a substrate of the EGFR kinase domain. The study therefore identifies many mechanistic details about this particular interaction that could be relevant from a clinical standpoint to develop better drugs that can reduce the oncogenic potential of these receptors. From a fundamental biology standpoint, the study reveals novel mechanisms of signal transduction at the membrane.

---

## [Decision Letter]

**Decision letter after peer review:**

Thank you for submitting your article "EGFR transactivates RON to drive oncogenic crosstalk" for consideration by *eLife*. Your article has been reviewed by 2 peer reviewers, one of whom is a member of our Board of Reviewing Editors, and the evaluation has been overseen by Jonathan Cooper as the Senior Editor. The reviewers have opted to remain anonymous.

The reviewers have discussed the reviews with one another and the Reviewing Editor has drafted this decision to help you prepare a revised submission.

Summary:

The study by Nitta et al., tries to understand the mechanisms behind the reports of crosstalk between the receptor tyrosine kinases – EGFR and RON. Earlier studies using immunoprecipitation showed that the receptors interact. This new paper provides evidence that the EGFR-RON crosstalk is unidirectional. They also show that this interaction takes place specifically at the membrane, rule out other intermediary molecules between EGFR and RON, and find that, in vitro, RON acts as a substrate of the EGFR kinase domain. The study therefore identifies many mechanistic details about this particular interaction that could be relevant from a clinical standpoint to develop better drugs that can reduce the oncogenic potential of these receptors. From a fundamental biology standpoint, the study reveals novel mechanisms of signal transduction at the membrane. Nevertheless, there are several major concerns.

Essential revisions:

1) The physiological relevance of the experimental system chosen to explore interactions between FGFR and RON is questionable. A431RON cells express approximately 2.2 million EGFR and approximately 92,000 RON receptors per cell. The HEKRON/EGFR cells express approximately 600,000 EGFR per cell and approximately 275,000 RON molecules per cell. The experimental system is therefore too artificial and too synthetic to reveal physiologically relevant quantitative insights for how EGF stimulation induce RON phosphorylation and how the dynamic properties of EGFR are linked to cellular responses that take place in a more physiologically relevant cellular contexts. Moreover, over-expression of EGFR and RON in these cells results in excessive sensitivity and responsiveness towards EGF or MSP stimulation which may not be detected in cells which express lower physiologically relevant levels of the two RTKs. In other words, over-expression may skew the true balance of the trans-phosphorylation reaction and yield flawed conclusions about the dynamic nature of the cellular system that is explored. Key results from this study should be verified in systems that have more physiological levels of the receptors. In particular: Is it possible to detect tyrosine phosphorylation of RON induced by EGF stimulation in cells expressing endogenous RON and EGFR which do not dramatically overexpress at least one of the two receptors ? Are RON and EGFR naturally expressed in the same cells or tissues or is their co-expression taking place only in certain cancer cells ?

2) Is transphosphorylation of RON by EGFR controlled by a tyrosine phosphatase? Does treatment of A431 or HEK cells a tyrosine phosphatase inhibitor also stimulate EGFR induced RON tyrosine phosphorylation ?

3) Colocalization and co-association:

(3a) Fluorescence and EM based-studies: Oligomerization of EGF-R has been a subject of many studies, and it appears from these works that at least a fraction of EGFR exists in oligomeric states in the absence of its ligand. Here the authors show large scale colocalization of RON with EGFR upon stimulation, however the EM-data show a significant fraction of RON- and EGFR co-exist in co-clusters at the resting state.

EM-based colocalization of RON and EGFR before and after stimulation by EGF is not properly quantified. The authors use Ripley's K-co-variate analysis and it is not clear what is being monitored in the lower panel in 1B. A more thorough analysis of the different modes present in this data would be necessary. Homo and hetero oligomer analysis at each time and concentration; EGFR-EGFR analysis, versus RON-RON as well as their co-variance, under all three conditions.

It would be important to understand how homogenous clusters of RON and EGFR change upon EGF and MSP stimulation, as the authors later in the manuscript find a cooperative action of MSP and EGF in activating RON. EM images in 1B quantify only the co-clustering. It will be especially interesting to see how RON homo-clusters change upon EGF stimulation.

Independently, it would be important to see how RON-EGFR co-clusters appear in the absence of ligand-induced activation at a larger spatial scale of the fluorescence based colocalization analysis. This could be achieved by labelling utilizing a fluorescently labelled EGFR and not detecting EGFR with a fluorescent probe that its ligand.

(3b) Immunoprecipitation studies: Separately, in the IP experiments shown in Supplementary Figure 1D, there is no evidence that addition of ligand induces co-association of RON and EGFR. This is in discordance with earlier studies. In fact, the data shown support the opposite; similar levels of co-IP of RON with EGFR with or without 50nM EGF. That phosphorylated RON co-IPs with EGFR after EGF addition is also evident, but this does not mean that kinase active EGFR is associating with RON after ligation. This must be discussed.

(3c) Endosomal Colocalization?:

RON-EGFR, RON-EAA and EGFR-EAA colocalization should be shown before and after activation with EGF, and quantified in order to draw conclusions. Figure 3C does not have a quantification of the colocalization indices that are essential to interpret these results. The authors are implying that the preferential association of EAA (in endosome) with EGFR as compared to RON might be an indication of the kinetics of the dissociation of RON and EGFR. This is not obvious from the data as one can still see a significant colocalization between RON and EGFR visually- presumably at the cell surface. This co-localization also needs to be better quantified, using more sophisticated analysis than simply 'visually looking' at correlated structures. It is also possible that there is an entirely different population of EGFR at the membrane that is being endocytosed that maybe never interacts with RON at all. Hence the conclusion does not follow from the given data.

4) Homo and hetero-oligomers:

Figure 4 clearly provides evidence for the existence of RON homo oligomers at the cell surface in the absence of a ligand, and hetero-dimers with EGF-ligated RON. The results show that EGFR and RON form transient complexes (Figure 4 D-F) and adding to the static picture from the EM data (when properly quantified), this experiment allows the authors to extract a k_off_ for the RON-EGFR interaction, but it is unclear if this k_off_ changes upon ligand activation by either MSP or EGF.

In the methods the authors state "For dual EGFR and RON tracking, cells were incubated with 200 pM 549 anti-HA-QD655 for 15 min at 37{degree sign}C, washed, and then stimulated with 50 pM QD605- 550 conjugated EGF-biotin". If the authors want to understand the interactions between 'activated' EGFR and RON, then it would be useful to introduce 50pM QD-EGF in the backdrop of 50 nM EGF, to visualize the role of activated EGFR. If the authors want a low labelling density to perform SPT, they should use unlabelled EGF at 50nM during the labelling. As the effect of the concentration of EGF is dose dependent, there might be larger changes in the membrane microenvironment caused by the EGFR activation (at 50nM EGF) which might change the rate constants the authors find for this interaction(at 50pM EGF). Ideally, the rate constant should be determined for the RON-EGFR complex in the presence and absence of ligand: a general criticism of this study.

5) Interaction with EGFR:

By citing evidence that receiver or acceptor mutants of EGFR (Zhang et al., 2006) are unable to facilitate RON activation, the authors claim that this is because RON cannot heterodimerize with EGFR monomers, and therefore it is necessary to invoke signalling from higher order clusters like heterotrimers etc. It is entirely possible that activated EGFR monomers which dissociate from activated EGFR dimers with an off rate of about 1.1 per second as determined from their own work (Coban et al., 2015) could be responsible for activating RON monomers or dimers. Though it is clear that EGFR and RON can homo-dimerize and that they can exist in higher order mixed clusters, it is not clear from the experiments in the manuscript whether it's the monomers or the dimers or higher order structures are responsible for this interaction, and hence potentiation of RON activation. However, once again the dimers are short lived and once activated and they may dissociate to form short-lived heterodimers with other membrane molecules, as observed with RON and EGF-liganded EGFR. The results provided only show that inactive EGFR cannot phosphorylate RON. This data is somewhat at odds with EGFR-Δ998 mutant which if anything potentiates RON much better than EGFR. Clearly this part of the manuscript requires a much better explanation.

6) General comments on statistics and data representation:

The authors have used bar plots throughout the manuscript, and this can lead to an inaccurate and incomplete representation of the data. As detailed here:

http://blogs.nature.com/methagora/2014/01/bring-on-the-box-plots-boxplotr.html

https://pagepiccinini.com/2016/02/23/boxplots-vs-barplots/

It would be best if the data is represented as box plots with the all the data points so that the entire distribution can be seen.

The red and green LUTs used in the manuscript might be hard for colour blind readers to understand and it would better if colour blind friendly LUT choices can be made.

7) Finally, the strong statements at the end of the discussion about the identification "of new mechanistic insights into therapeutic resistance" are not warranted and should be toned down.

---

## [Author Response]

Essential revisions:1) The physiological relevance of the experimental system chosen to explore interactions between FGFR and RON is questionable. A431RON cells express approximately 2.2 million EGFR and approximately 92,000 RON receptors per cell. The HEKRON/EGFR cells express approximately 600,000 EGFR per cell and approximately 275,000 RON molecules per cell. The experimental system is therefore too artificial and too synthetic to reveal physiologically relevant quantitative insights for how EGF stimulation induce RON phosphorylation and how the dynamic properties of EGFR are linked to cellular responses that take place in a more physiologically relevant cellular contexts. Moreover, over-expression of EGFR and RON in these cells results in excessive sensitivity and responsiveness towards EGF or MSP stimulation which may not be detected in cells which express lower physiologically relevant levels of the two RTKs. In other words, over-expression may skew the true balance of the trans-phosphorylation reaction and yield flawed conclusions about the dynamic nature of the cellular system that is explored.

We would like to emphasize that the goal of this work was to study RON/EGFR crosstalk in the context of cancers where receptor tyrosine kinases are frequently overexpressed and can number in the millions of receptors per cell. RON and EGFR are indeed found to be co-overexpressed and linked to poor outcomes or resistance to targeted therapies in a number of tumor types including, colon (Graves-Deal 2019 PMC6407678), head and neck (Keller, 2013 PMC3721396) and bladder cancers (Hsu 2006 PMID: 17070309). Previous work in this area that we build upon has also focused on the cancer setting and has predominantly relied on cell lines with receptor overexpression to interrogate RON-EGFR interactions (Hsu 2006 PMID: 17070309; Peace 2003 PMID: 14499632). To ensure that the crosstalk was not an artifact of a single cell type, we studied two different cell lines (A431 and HEK-293) with different expression levels/ratios. Notably, RON isoforms or EGFR oncogenic mutants, which may be present in cell lines endogenously expressing these receptors, have also been shown to play roles in cancer and could influence RON/EGFR interactions (Krishnaswamy 2015 PMC4723846; Krishnaswamy 2017 PMC5679369). We chose to focus here on the interaction of full-length wildtype receptors. Therefore, while our work does not directly address the question of how RON and EGFR interact under normal physiological levels, we do provide support for the ability of RON and EGFR to interact in cancer systems, and importantly, we provide new insight into the molecular mechanisms of this interaction.

Key results from this study should be verified in systems that have more physiological levels of the receptors. In particular: Is it possible to detect tyrosine phosphorylation of RON induced by EGF stimulation in cells expressing endogenous RON and EGFR which do not dramatically overexpress at least one of the two receptors ? Are RON and EGFR naturally expressed in the same cells or tissues or is their co-expression taking place only in certain cancer cells ?

The reviewer raises an interesting and important point. RON and EGFR are both found in epithelial cells including skin, colon, bladder, breast and lung (Sakamoto 1997 PMID: 9045873, Wang 2007 PMID: 17955509, Chen 2016 PMID: 33003261), so there is potential for interactions under normal physiological conditions. We agree with the reviewer that it would be important to explore RON/EGFR interactions under more physiological conditions. We attempted to pursue this suggestion, however, as described below, the results were inconsistent.

We first identified several cell lines that endogenously co-express EGFR and RON at low/more physiological levels, based on RNA expression from the Human Protein Atlas database. These included MCF7, A549, HEK, HCT116 and SKBR3. Of these, we only found HCT and SKBR3 to show detectable expression of EGFR and RON by western blot (example blots shown in Author response image 1). Immunoprecipitation of RON with an anti-RON antibody revealed a phosphotyrosine (pY99) band in the unstimulated HCT116 cells. However, this band was only visible in about half of the blots and when visible its intensity was *reduced* with ligand stimulation (Author response image 1) . This suggested to us that the band may not truly be related to RON. By using two-color Western Blot analysis, it was made clear that the pY99 band was not aligned with pro-RON (Author response image 1) , since it was located just below the total RON bands. This result highlights the importance of our two-color western blot approach used throughout the manuscript, where all phosphorylation was confirmed to overlay with the proper total protein band. It also demonstrates that detection of endogenous/low-level RON protein will be complicated given the current available reagents and requires more time to follow up thoroughly.

**Author response image 1. sa2fig1:** Detection of RON and EGFR in human cancer cell lines with low, endogenous expression of both receptors. HCT116 colorectal carcinoma cells, SKBR3 breast cancer cells and A549 lung carcinoma cells were cultured overnight and then stimulated with 50 nM EGF or 5 nM MSP for 5 min. (A) RON receptor was IP’d with an anti-RON antibody and phosphorylated proteins in the IP were detected using a pY99 antibody. Here, detection was with HRP-antibodies. (B) Total cell lysates were run on a western blot and phosphoEGFR (pY1068) and total EGFR were detected. (C) Two-color detection using fluorescently-labeled antibodies for western using HCT116 cells. The phosphorylated protein does not overlap with full-length RON in the IP samples. The phosphorylated protein could be a truncated RON isoform or a different protein that co-Ips with RON in these cells. (D) The phospho-EGFR band does overlap with the full-length EGFR band.

While it was difficult to detect endogenous RON phosphorylation in these cells, the detection of EGFR phosphorylation was consistent and further confirms that MSP does not lead to EGFR phosphorylation under these cellular conditions Author response image 1. We feel that understanding EGFR/RON crosstalk at physiological expression levels, while interesting and important, is beyond the scope of this paper. Determining the similarities or differences in physiological versus oncogenic signaling does not preclude the results of our paper or the previous work showing crosstalk in the oncogenic/overexpression context.

2) Is transphosphorylation of RON by EGFR controlled by a tyrosine phosphatase? Does treatment of A431 or HEK cells a tyrosine phosphatase inhibitor also stimulate EGFR induced RON tyrosine phosphorylation ?

We believe that the reviewers are asking whether crosstalk could be due to an indirect mechanism where an EGFR-controlled phosphatase is regulating RON. For example, recruitment of a phosphatase to EGFR could remove it from RON, allowing for RON phosphorylation. In general, all phosphorylation is regulated by phosphatases and phosphatase inhibition will result in increased phosphorylation of both RON and EGFR. However, we would not be able to parse out the difference between EGFR-controlled phosphatases and general phosphatase activity. Therefore, this experiment would not be able to directly identify an EGFR-controlled phosphatase regulation of RON. While it is possible that phosphatases are playing a role, our in vitro kinase assay (Figure 6C) shows that phosphatases are not required because the EGFR kinase domain alone can phosphorylate the RON tail.

We note also that in Figure 6—figure supplement 2, we show that RON has similar dephosphorylation kinetics whether transactivated by WT EGFR or EGFR-Δ998 (that lacks the major sites for cytosolic protein recruitment and signal propagation). If EGFR signaling was needed to recruit or activate phosphatases, we would have expected RON dephosphorylation to be altered with EGFR-Δ998, but no difference was seen.

(3) Colocalization and co-association:(3a) Fluorescence and EM based-studies: Oligomerization of EGF-R has been a subject of many studies, and it appears from these works that at least a fraction of EGFR exists in oligomeric states in the absence of its ligand. Here the authors show large scale colocalization of RON with EGFR upon stimulation, however the EM-data show a significant fraction of RON- and EGFR co-exist in co-clusters at the resting state.

In this study, we have used confocal and electron microscopy to demonstrate the potential for RON and EGFR interactions. Due to their limitations, both methods suggest, but neither prove, the existence of homo- or hetero-oligomers. For example, confocal microscopy is limited by a resolution of ~250 nm. Therefore, while two distinctly labeled receptors may show colocalization, this does not confirm interactions, since proteins are ~ a factor of 10 smaller than this diffraction limit. That being said, if no colocalization is seen, then it would strongly suggest a low probability of interaction. In a similar manner, EM can provide higher resolution evidence of protein proximity, indicating clustering or exclusion. However, observation of protein (co-) clustering does not confirm physical oligomerization/interactions. It is well appreciated that the membrane can form domains (actin corrals, lipid rafts, protein islands) that can organize proteins into clusters, even in the resting state, without the proteins themselves directly interacting. Therefore, while RON and EGFR appear to homo- and hetero-cluster in the EM images, we cannot use this as proof of oligomerization. The intent of these data was to determine, as a first step, whether the two receptors are within close enough proximity to potentially interact. We turned to other methods, in particular two-color SPT, to detect the direct interactions of receptors.

EM-based colocalization of RON and EGFR before and after stimulation by EGF is not properly quantified. The authors use Ripley's K-co-variate analysis and it is not clear what is being monitored in the lower panel in 1B.

We thank the reviewers for pointing out our insufficient description of the EM analysis. The method presented is the Ripley’s K bivariate test (also called cross-type Ripley’s K Function) that is used to define the spatial relationship of two different sizes of gold particles (i.e., two different protein species). In the analysis presented in (revised) Figure 2B, the experimental values for L(r)-r (red line) is plotted as a function of distance (r). The dashed lines represent the values of L(r) for complete spatial randomness with a 99% confidence envelope (estimated from 100 Monte Carlo simulations for each ROI analyzed). When the experimental values fall above the confidence interval, the deviation from complete spatial randomness is statistically significant and indicates that the two labels show co-clustering. This has been used by our group in a number of previous publications (Wilson 2004 PMC420084; Yang et al., JCS 2007 PMID 17652160) and is distinct from Ripley’s K test that determines L(r)-r for a single species. We have expanded the description in the figure legend and methods to make this clearer.

We have added the following to the legend of Figure 2:

“Ripley’s K bivariant function was used to evaluate co-clustering. The experimental values for L(r)-r (corresponding to EM image directly above) are shown in magenta and the 99% confidence window for complete spatial randomness is plotted as dashed lines. In each case, experimental values are seen to fall above the confidence window, indicating co-clustering.”

As well as adding the following to the Methods (p. 32):

“Ripley's bivariate K test was used to determine if co-clustering of species is significant (Wilson et al., 2004; Yang et al., 2007), with a critical interaction distance of 50 nm. Data within the confidence window are not significantly co-clustered. When the experimental values are found above the confidence window the deviation from complete spatial randomness is statistically significant and indicates that the two labels are co-clustering.”

A more thorough analysis of the different modes present in this data would be necessary. Homo and hetero oligomer analysis at each time and concentration; EGFR-EGFR analysis, versus RON-RON as well as their co-variance, under all three conditions.It would be important to understand how homogenous clusters of RON and EGFR change upon EGF and MSP stimulation, as the authors later in the manuscript find a cooperative action of MSP and EGF in activating RON. EM images in 1B quantify only the co-clustering. It will be especially interesting to see how RON homo-clusters change upon EGF stimulation.

We agree that further study of the lateral organization of RON and EGFR under a range of activation conditions would be of interest. However, a full examination of all possible combinations is not needed for the current manuscript’s focus on understanding whether EGFR and RON are capable of undergoing direct interactions. We also realize that the initial order of the figures may have been distracting, since we showed a focus on EGF stimulation of cells, before we explained that the crosstalk only occurs with EGF and not MSP stimulation. We have re-ordered the manuscript so that biochemical analysis of the crosstalk is the first figure, followed by confocal and EM imaging.

We have included additional analysis of the existing EM data in Figure 2—figure supplement 2. We provide a larger view of an EM sheet for each condition, along with the corresponding Hopkin’s statistical test for homoclustering. The Hopkin’s statistic confirms that each receptor is already found to homocluster in the resting state. We also provide mean statistics across multiple images for Hopkin’s, percent of receptors found in homoclusters and percent of receptors found in coclusters. We do not observe a significant change in these quantities as a function of EGF activation, however the clustering/coclustering is quite high to start with in the A431^RON^ cells. Future studies would be better performed on cells with a lower expression level in order to determine if subtle changes could be detected. This, however, requires generation of new cell lines and is ongoing work.

Independently, it would be important to see how RON-EGFR co-clusters appear in the absence of ligand-induced activation at a larger spatial scale of the fluorescence based colocalization analysis. This could be achieved by labelling utilizing a fluorescently labelled EGFR and not detecting EGFR with a fluorescent probe that its ligand.

We thank the reviewers for the suggestion to expand the colocalization analysis. We have now included further immunofluorescence experiments and analysis comparing RON colocalization with either fluorescent EGF or an anti-EGFR antibody (Figure 2—figure supplement Figure 1). Quantification of colocalization for the confocal images is also included. This shows that RON is colocalized with both resting (unliganded) EGFR and EGF-bound EGFR on the plasma membrane. This is consistent with the observation in EM that shows co-clustering in both resting and EGF-activated cells.

(3b) Immunoprecipitation studies: Separately, in the IP experiments shown in Supplementary Figure 1D, there is no evidence that addition of ligand induces co-association of RON and EGFR. This is in discordance with earlier studies. In fact, the data shown support the opposite; similar levels of co-IP of RON with EGFR with or without 50nM EGF. That phosphorylated RON co-IPs with EGFR after EGF addition is also evident, but this does not mean that kinase active EGFR is associating with RON after ligation. This must be discussed.

In our results, we report that EGFR can co-IP with RON even in the absence of ligand. This is consistent with earlier studies (Peace 2003 PMID: 14499632, Hsu 2006 PMID: 17070309) where they also show co-IP in the absence of ligand. Where there is some difference, is in the fact that we do not see a consistent increase in EGFR pulldown with RON after ligand stimulation. However, there are significant differences in our approaches that likely account for this. Peace et al. does not emphasize an increase with ligand, but does highlight the pre-association in the absence of ligand (Peace 2003, Figures 4 and 5), as we have also seen. Hsu suggests an increase in co-IP between RON and EGFR with MSP stimulation (Figure 1A of Hsu 2006), yet only a single blot is shown and the blots are not quantified. It is also important to point out that these blots used the strip-and-reprobe approach, while we used simultaneous two-color imaging that removes potential stripping artifacts. Furthermore, both of these studies performed co-IPs at 30 min post-stimulation. We are looking at early events and perform most biochemical experiments at 5 min post-stimulation. Finally, we note that we could not always see evidence of co-IP, suggesting a weak interaction. We have modified the text on p. 7 to highlight these points:

“EGFR was often detected in RON immunoprecipitates, in both resting and stimulated cells, as a band co-migrating with pro-RON at 180 kDa via western blot analysis (using EGFR or EGFRPY1068 antibodies) or identified by mass spectrometry, (Figure 1 – Supplementary Figure 1D and Supplementary Table 1). Co-immunoprecipitation of RON and EGFR in unstimulated cells has been reported previously (Hsu et al., 2006; Peace et al., 2003). In contrast to that previous work, we do not observe an increase in co-precipitation with ligand stimulation. However, we note that co-IP was not always evident, suggesting weak interactions, and our experiments were performed at earlier time points (5 min) than the previous studies (30 min).”

We have mentioned in the discussion that the dynamics of the EGFR-RON interaction in the absence of ligand remain to be determined. However, even without that information, it is clear from our data that crosstalk occurs via EGFR interacting with RON such that the EGFR kinase can phosphorylate RON.

(3c) Endosomal Colocalization?:RON-EGFR, RON-EAA and EGFR-EAA colocalization should be shown before and after activation with EGF, and quantified in order to draw conclusions. Figure 3C does not have a quantification of the colocalization indices that are essential to interpret these results. The authors are implying that the preferential association of EAA (in endosome) with EGFR as compared to RON might be an indication of the kinetics of the dissociation of RON and EGFR. This is not obvious from the data as one can still see a significant colocalization between RON and EGFR visually- presumably at the cell surface. This co-localization also needs to be better quantified, using more sophisticated analysis than simply 'visually looking' at correlated structures. It is also possible that there is an entirely different population of EGFR at the membrane that is being endocytosed that maybe never interacts with RON at all. Hence the conclusion does not follow from the given data.

Thank you for this suggestion. We have now confirmed the lack of RON found in early endosomes with colocalization analysis (Figure 3—figure supplement 2). We have also performed additional experiments to examine EGFR and RON localization using a second labeling method (Figure 2D). The conclusion that we take from the results in Figure 3 (including the SPT and EM) is that productive EGFR and RON interactions occur soon after EGF addition – while both receptors are still at the plasma membrane. This was important to determine since earlier studies (Hsu, et al.; Peace et al.) were performed at 30 min post-stimulation, a time at which most stimulated receptors would be expected to have endocytosed, so location of crosstalk was not clear. It is intriguing to consider, as suggested by the reviewer, that EGFR bound to RON is held longer on the membrane. We have expanded the text on p. 12 to indicate this possibility:

“These results suggest that EGFR/RON interactions are either sufficiently transient that EGFR is sorted for endocytosis, while RON remains on the surface, or that EGFR complexed with RON is retained longer on the cell surface.”

We assert that either way our conclusion is not changed: EGFR and RON interact at the plasma membrane and signaling from endosomes is not required.

4) Homo and hetero-oligomers:Figure 4 clearly provides evidence for the existence of RON homo oligomers at the cell surface in the absence of a ligand, and hetero-dimers with EGF-ligated RON. The results show that EGFR and RON form transient complexes (Figure 4 D-F) and adding to the static picture from the EM data (when properly quantified), this experiment allows the authors to extract a k_off_ for the RON-EGFR interaction, but it is unclear if this k_off_ changes upon ligand activation by either MSP or EGF.In the methods the authors state "For dual EGFR and RON tracking, cells were incubated with 200 pM 549 anti-HA-QD655 for 15 min at 37{degree sign}C, washed, and then stimulated with 50 pM QD605- 550 conjugated EGF-biotin". If the authors want to understand the interactions between 'activated' EGFR and RON, then it would be useful to introduce 50pM QD-EGF in the backdrop of 50 nM EGF, to visualize the role of activated EGFR. If the authors want a low labelling density to perform SPT, they should use unlabelled EGF at 50nM during the labelling. As the effect of the concentration of EGF is dose dependent, there might be larger changes in the membrane microenvironment caused by the EGFR activation (at 50nM EGF) which might change the rate constants the authors find for this interaction(at 50pM EGF). Ideally, the rate constant should be determined for the RON-EGFR complex in the presence and absence of ligand: a general criticism of this study.

We have focused this manuscript on EGF-bound EGFR because this is the condition where crosstalk was observed. While adding to the overall picture, quantifying interactions between unliganded EGFR and RON is not necessary to the new insights already provided here: EGFRRON undergo *bona fide* interactions under the conditions where signaling is happening (ie, the presence of EGF). We do agree that this is important information to continue to pursue and such work is ongoing in the lab. We are working on developing orthogonal labeling methods (including QD-MSP) to allow for capturing of EGFR and RON interactions in the absence of EGF or in response to MSP. One could imagine a large number of experiments to exhaustively quantify RON-RON, RON-EGFR and EGFR-EGFR interactions across multiple ligands and doses. This would warrant a separate publication. Showing the direct interaction between EGF-bound EGFR and unliganded RON (the situation where crosstalk occurs), as we have done here, is a critical step forward in our mechanistic understanding of crosstalk.

We understand the experiment to use a high dose dark EGF that is being suggested. However, there are difficulties in performing and interpreting the proposed experiments. That high amount of ligand activation leads to rapid cell morphology changes and endocytosis of EGFR that makes capturing dimer events on the cell surface difficult. Also, the saturation of all receptors with ligand, the majority of which are dark, make it much less probable that two QD-labeled receptors will find each other and form dimers when they have an abundance of dark EGF-EGFR with which to interact. Therefore, we are not confident in the interpretation of this approach and prefer to follow up with new technologies currently under development to address this sort of question.

5) Interaction with EGFR:By citing evidence that receiver or acceptor mutants of EGFR (Zhang et al., 2006) are unable to facilitate RON activation, the authors claim that this is because RON cannot heterodimerize with EGFR monomers, and therefore it is necessary to invoke signalling from higher order clusters like heterotrimers etc. It is entirely possible that activated EGFR monomers which dissociate from activated EGFR dimers with an off rate of about 1.1 per second as determined from their own work (Coban et al., 2015) could be responsible for activating RON monomers or dimers. Though it is clear that EGFR and RON can homo-dimerize and that they can exist in higher order mixed clusters, it is not clear from the experiments in the manuscript whether it's the monomers or the dimers or higher order structures are responsible for this interaction, and hence potentiation of RON activation. However, once again the dimers are short lived and once activated and they may dissociate to form short-lived heterodimers with other membrane molecules, as observed with RON and EGF-liganded EGFR. The results provided only show that inactive EGFR cannot phosphorylate RON. This data is somewhat at odds with EGFR-Δ998 mutant which if anything potentiates RON much better than EGFR. Clearly this part of the manuscript requires a much better explanation.

We thank the reviewer for pointing out our limited interpretation of the results. Crosstalk certainly could be happening as described, where an EGFR dimer forms, activates and dissociates to then interact with RON.

We have modified the discussion on p. 26 to reflect this possibility:

“We found that RON cannot serve as an activator or receiver kinase in an EGFR/RON heterodimer. Instead, formation of a signaling-competent EGFR homodimer appears to be first required to initiate EGF-driven RON phosphorylation. Considering that RON homo-interactions were observed by two-color SPT in both resting and liganded states, we postulate that the heterocomplex consists of a RON dimer interacting with a ligand-bound EGFR dimer or EGFR monomer that has recently dissociated from a homodimer after kinase activation.”

Either way, it seems that RON cannot act as the activator to EGFR. Therefore, RON cannot simply substitute for an activator or receiver in a possible EGFR-RON pair.

This data is not necessarily at odds with the *EGFR-Δ998*, as that mutant retains an active kinase. We have added discussion about how the removal of the EGFR tail might augment RON phosphorylation (p. 26):

“It is also conceivable that adaptor proteins recruited to the EGFR tail (Biscardi et al., 1999; Yamauchi et al., 1998) could subsequently phosphorylate RON. However, we found that neither inhibition of c-Src activity nor removal of the EGFR cytoplasmic tail (EGFR-Δ998) prevented crosstalk with RON. Adaptor proteins may explain the enhanced phosphorylation of RON that was seen in cells expressing EGFR-Δ998. For instance, Grb2 has been reported to inhibit RON autophosphorylation, raising the possibility that loss of Grb2 recruitment by EGFR-Δ998 could reduce local Grb2 concentration and increase RON phosphorylation. Alternatively, removal of the EGFR C-terminal tail diminishes the recruitment of downstream EGFR substrates, limiting substrate competition and making the RON C-terminal tail the preferred substrate in the heterooligomeric complexes. Together, along with the identification of RON as a substrate for EGFR kinase, our results establish that crosstalk is mediated by receptor-receptor interactions.”

6) General comments on statistics and data representation:The authors have used bar plots throughout the manuscript, and this can lead to an inaccurate and incomplete representation of the data. As detailed here:http://blogs.nature.com/methagora/2014/01/bring-on-the-box-plots-boxplotr.htmlhttps://pagepiccinini.com/2016/02/23/boxplots-vs-barplots/It would be best if the data is represented as box plots with the all the data points so that the entire distribution can be seen.The red and green LUTs used in the manuscript might be hard for colour blind readers to understand and it would better if colour blind friendly LUT choices can be made.

We thank the reviewers for these excellent suggestions. All bar graphs have been reformatted to include individual data points overlayed with the mean +/- SD.

The images have been changed to color blind friendly LUT throughout.

7) Finally, the strong statements at the end of the discussion about the identification "of new mechanistic insights into therapeutic resistance" are not warranted and should be toned down.

We have toned down this section, modifying the discussion on p. 26-27:

“While further study is needed to determine the exact stoichiometry of the EGFR/RON complex, the molecular mechanisms governing EGFR/RON crosstalk described here suggest that disruption of interactions between RON and EGFR could provide a therapeutic advantage. In particular, we have shown that RON signaling outcomes can be stimulated by EGFR in the absence of MSP and even if RON kinase activity is inhibited.”